

# Depth Effects of Long-term Organic Residue Application on Soil Organic Carbon Stocks in Central Kenya

Claude Raoul Müller[1], Johan Six[1], Daniel Mugendi Njiru[2], Bernard Vanlauwe[3], Marijn Van de Broek[1]

[1] Department of Environmental Systems Science, ETH Zurich, 8092 Zürich, Switzerland
[2] Department of Land and Water Management, University of Embu, P.O. Box 6-60100, Embu, Kenya
[3] International Institute of Tropical Agriculture (IITA), c/o ICIPE Compound, P. O. Box 30772–00100, Nairobi, Kenya

*Correspondence to*: Claude Raoul Müller (claude.mueller@usys.ethz.ch)

## Abstract

In arable soils, a substantial portion of soil organic carbon (SOC) is stored below the plough layer. To develop sustainable soil management strategies, it is important to assess how they affect the quantity of SOC stored in the subsoil. Therefore, we investigated the impact of organic and inorganic nutrient inputs on SOC stocks down to 70 cm depth in a long-term field trial in Embu, Kenya. There were 3 organic input treatments (manure, *Tithonia diversifolia* residues, and maize stover) and a control treatment, each with and without the application of mineral nitrogen. These different treatments were applied to a maize monoculture over 38 growing seasons (19 years). Our results show that manure application had the largest positive impact on SOC stocks compared to the control, which was observed down to 60 cm depth. In contrast, *Tithonia diversifolia* and maize stover significantly increased SOC compared to the control only within the top 20 cm and 40 cm, respectively. Among the three organic residue treatments, only the application of manure had a significant effect on the SOC stock of the subsoil (i.e., the 30-70 cm depth layer). However, when considering the whole measured profile (i.e., 0-70 cm), all treatments led to significantly higher SOC stocks compared to the 91 ± 12 t C ha⁻¹ of the control: manure had the highest stocks (120 ± 24 t C ha⁻¹), followed by maize stover (112 ± 17 t C ha⁻¹) and *tithonia diversifolia* (105 ± 11 C t ha⁻¹). Mineral nitrogen application did not have a significant impact on SOC stocks down to 70 cm depth. Overall, our findings indicate that the subsoil in the studied agroecosystems is affected by the type of added organic amendments. Our results imply that gathering knowledge on the soil below the typically studied 0-30 cm depth layer will improve the overall assessment of agroecosystem properties, which is necessary to optimize soil system resilience, limit organic matter losses and improve crop productivity.



## 1 Introduction

Globally, food production systems need to satisfy the increasing food demand of an ever-more demanding and growing population. However, changing land use for food production and increasing productivity leads to intense use of our limited natural resources, which can have a negative impact on the quality of groundwater and the atmosphere. In the long term, it also leads to soil quality degradation (e.g., through crusting, compaction, erosion, and depletion of organic matter) (Lal, 2008). Tropical regions are currently experiencing the most significant land-use change globally, largely due to the rapid conversion of natural habitats into agricultural land (Hansen et al., 2013). This swift alteration of land use has substantial effects on the amount of soil organic carbon (SOC) stored in tropical ecosystems, which contains approximately 44% of the global SOC (Veldkamp et al., 2020). The concentration of SOC is an important component of soil fertility, as it serves as a vital nutrient source for plants and microbes (Chen and Aviad, 1990; Trevisan et al., 2010). Thus, maintaining SOC levels over time is crucial for sustaining soil productivity, especially in the highly weathered soils of tropical regions, which are particularly prone to OC loss (Feller and Beare, 1997). This is clear from the fact that the conversion of tropical forests to agriculture typically results in an average reduction of organic carbon (OC) concentration in the topsoil (down to a depth of 30 cm) of around 50% within 25 years (Veldkamp et al., 2020).

In general, the regular harvesting of crops in farming systems results in the removal of nutrients that the plants have taken up. Thus, inadequate inputs of organic matter and nutrients in farming systems of sub-Saharan Africa (SSA) often leads to the loss of SOC and nutrient depletion, contributing to the lower yields observed in SSA compared to other regions of the world (Vanlauwe and Giller, 2006). Therefore, agricultural management should not only aim at increasing crop yield in the short term, but also at maintaining SOC at a level that maintains the key function that it regulates in the long term. Ideally, the management of cropping systems should respect the principles of sustainable intensification by maximizing production while preserving the environment; different management approaches exist to reach this goal (Pretty, 2011). A good example is integrated soil fertility management (ISFM) which is defined by Vanlauwe et al. (2010) as *"a set of soil fertility management practices that necessarily include the use of fertilizer, organic inputs, and improved germplasm combined with the knowledge on how to adapt these practices to local conditions, aiming at maximizing agronomic use efficiency of the applied nutrients and improving crop productivity. All inputs need to be managed following sound agronomic principles"*. This approach promotes various practices, including the combined application of mineral fertilizer and organic material such as plant residues, manure, and compost (i.e., organic residues). Such practices aim to enhance crop productivity while preserving soil fertility, especially in a context where input availability is often limited, as is the case for smallholder farming in SSA.

Although the turnover rate of SOC is faster under tropical than temperate climate (Wang et al., 2018), applying organic residues over multiple years in tropical croplands has the potential to significantly increase the SOC content (Fujisaki et al., 2018). However, its contribution to the amount of SOC is, in part, controlled by the type of added organic residues (Córdova et al.,



2018). The residue quality has traditionally been defined by the nitrogen, lignin, and polyphenol content of organic inputs; overall, the lower the (lignin + polyphenol)/N ratio, the higher the quality of the residue (Chivenge et al., 2011b; Vanlauwe et al., 2005). According to Córdova et al. (2018), applying high-quality residues has the potential to lead to a higher accumulation of stable C in the soil than low-quality residues. Nevertheless, an incubation experiment on tropical soil indicated that, while this might be true in the short term, initial OR quality does not affect SOC accumulation in the long term (Gentile et al., 2011). Moreover, Laub et al. (2023) found that for four tropical agroecosystems in Kenya, it is mainly the application of animal manure that has the potential to limit SOC losses from intensive arable soil use. Most importantly, according to Vanlauwe et al. (2015), the efficiency of strategies to increase the SOC content is determined by variations in local soil conditions. This is supported by the variability in results obtained by different studies in tropical soils that either suggest that increasing the SOC content with adapted practices is possible (Adams et al., 2020; Fujisaki et al., 2018; Laub et al., 2022) or the application of large amounts of organic residues is not sufficient to increase or even maintain SOC content (Kihara et al., 2020; Cardinael et al., 2022; Laub et al., 2023b). Different factors, such as initially high SOC contents, favorable conditions to decomposition, and the limited capacity of 1:1 kaolinite clay minerals to stabilize OC, contribute to consistent SOC losses despite the application of organic residues in tropical soils (Laub et al., 2023b; Six et al., 2002; Sommer et al., 2018).

In most studies on agroecosystems, only the topsoil (i.e., down to 30 cm depth) is considered, while the effect of organic residue application on the soil below the plough layer is rarely studied (Yost and Hartemink, 2020). However, while field studies on plant nutrient acquisition from the subsoil are rare, it was shown that crop nutrient availability could be improved when organic residues and mineral fertilizer were placed in deeper soil layers (20-40 cm depth), showing that subsoil OC can play an active role in nutrient use efficiency (Liu et al., 2021; Ma et al., 2022). Also, studying only topsoil OC is not sufficient to estimate the effect of management practices on the OC storage capacity of soils because it is estimated that 50% of SOC stocks are located below 30 cm (Balesdent et al., 2018; Lal, 2018; Shumba et al., 2024; Yost and Hartemink, 2020). Moreover, not only do plant roots and their associated exudates provide direct OC inputs to the subsoil (Van De Broek et al., 2020), also organic matter applied in the topsoil undergoes cycles of sorption and desorption through microbial processing, gradually migrating down the soil profile (Kaiser and Kalbitz, 2012). Finally, while the average residence time of subsoil OC ranges from decades to millennia (Balesdent et al., 2018; Mathieu et al., 2015), making subsoil potentially a long-term carbon sink (Rumpel and Kögel-Knabner, 2011), only a small proportion of new carbon inputs traverse the soil profile and can potentially be stabilized in the long-term (Sierra et al., 2024).

Stable isotopes of carbon, specifically $\delta^{13}C$ and $\delta^{14}C$, are useful indicators for SOM cycling in agricultural soils. The $\delta^{13}C$ reflects the stable carbon isotopic composition in plant tissues, which can vary according to the different plant photosynthesis (i.e., C3 or C4 photosynthesis). Notably, C3 plants (e.g. *Tithonia diversifolia*) discriminate more against $^{13}C$ compared to C4 plants (e.g., maize) and have therefore a typical value around -28‰, while C4 plants have a typical $\delta^{13}C$ value of -12‰ (Balesdent et al., 1987; Farquhar G D et al., 1989). Nevertheless, these typical values can vary and for tropical vegetation the





rang is large and often lower than 28‰ (Kohn, 2010; Martinelli et al., 2021). The $\delta^{14}C$ can be used as a proxy for the age of carbon-containing soil components, providing insights into the turnover times of different soil organic matter pools, which can be particularly important for understanding the long-term sequestration and storage of carbon in soils (Ehleringer et al., 2000; Trumbore et al., 1989).

Given the lack of knowledge related to how organic amendments and fertilizer management affect subsoil organic matter dynamics, our study aims to examine the impact of different combinations of organic matter inputs and mineral nitrogen fertilizer on SOC stocks down to 70 cm by answering the following research questions:

1    How do (1) organic residue application and (2) mineral nitrogen fertilization affect the SOC stock along the depth profile down to 70 cm?

2    Which organic residue addition leads to the largest stocks of OC along the soil profile?

## 2 Methods

### 2.1 Study design

#### 2.1.1 Study area

The field trial was initiated in 2002 in Embu (Central Kenya: 0°30′ S, 37°27′ E; 1380 m above sea level). It is characterized by a continuous maize monoculture with two annual growing seasons that coincide with the bimodal precipitation pattern, with rainy seasons generally occurring from March to September (the long rainy season) and October to February (the short rainy season). The average annual precipitation is 1175mm, and the mean annual temperature is 20.1°C. The soil in this area is a highly weathered Humic Nitisol. Originally, the site was covered by tropical evergreen forest before having been converted into a low-intensity agricultural area. Before the experiment started, the site was terraced to ensure that the plots were level. Additional details about the site can be found in previously published studies (Chivenge et al., 2009, 2011a; Gentile et al., 2008; Laub et al., 2023b)

In a recent study conducted at this site, Laub et al., (2023b) showed that all nutrient management treatments at Embu resulted in significant SOC losses, but the use of farmyard manure was the most effective strategy to minimize the loss of SOC. In their study on Embu site, Laub et al., (2023b) observed consistent decreases in soil organic carbon (SOC) contents over time for all treatments (Fig. S8). The control treatment showed the greatest declines, with a loss of about 0.6-0.7 g C kg$^{-1}$ yr$^{-1}$. Despite differences in treatment effectiveness in maintaining OC, farmyard manure (4 t C ha$^{-1}$) emerged as the most effective strategy to limit SOC losses at Embu with the lowest OC loss (i.e. 0.4 g C kg$^{-1}$ yr$^{-1}$). However, their focus was on the effects of treatments above 15 cm depths.



### 2.1.2 Experimental design

130

The field experiment has a split-plot design with three replications. Four nutrient management treatments were selected for the present study, involving the application of three organic residues with varying qualities (at a rate of 4.0 t C ha$^{-1}$ yr$^{-1}$) and a control treatment (*Control*; no organic residues were added). These organic resources were applied once annually before the long rainy season. Organic amendments were incorporated in the soil through manual tilling to a depth of approximately 15

135 cm. Each plot was further divided into two subplots, with one subplot receiving an application of 120 kg mineral N ha$^{-1}$ per growing season (*+N* treatment), while the other remained devoid of any mineral nitrogen input (*-N* treatment). Mineral N (CaNH$_4$NO$_3$) was applied twice during each growing season: an initial 40 kg N ha$^{-1}$ at planting and the remaining 80 kg N ha$^{-1}$ approximately six weeks later as a top dressing. In every growing season, all plots were subjected to a uniform application of 60 kg P ha$^{-1}$ in the form of triple superphosphate and 60 kg K ha$^{-1}$ as muriate of potash at planting. The organic resources

140 utilized in the present study exhibited variations in quality, characterized by differences in nitrogen, lignin, and polyphenol contents. These included pruned leaves, including stems with a thickness of less than 2 cm, from *Tithonia diversifolia* (*Tithonia*; high quality with rapid turnover), stover of *Zea mays* (*Stover*; low quality and fast turnover), and locally available farmyard manure (*Manure*; intermediate to high quality with intermediate turnover). The relevant properties of the different organic inputs are given in (Table 1). After harvest, any remaining parts of the maize plant were removed from the plots.

145 Therefore, the only carbon inputs left from the maize crop came from roots and root exudates.

This study focuses on the effect of organic residues as compared to the control. The treatments include both the *-N* and *+N* alternatives combined together, and are referred to here as *Manure* (i.e., farmyard manure), *Tithonia* (i.e., residues of *Tithonia diversifolia*), *Stover* (residues of maize stover), and *Control* (i.e., no organic residue added). Whenever we refer to a treatment specifically with or without mineral N, a *-N* or a *+N* symbol is added after the treatment name. For example, *Manure-N*

150 represents application of manure without mineral N, *Control+N* means no that only mineral fertilizer was added, while *Tihtonia* includes both *Tithonia+N* and *Tithonia-N* treatments.

Table 1: Mean dry matter properties of the organic resources applied (source: Laub et al., 2023b)

| Measured property | Manure | Stover | *Tithonia* |
|---|---|---|---|
| C (g kg$^{-1}$) | 234 | 397 | 345 |
| N (g kg$^{-1}$) | 18.1 | 7.2 | 33.2 |
| C:N ratio | 12.3 | 58.7 | 12.4 |
| P (g kg$^{-1}$) | 3.1 | 0.4 | 2.3 |
| K (g kg$^{-1}$) | 19.4 | 9 | 37.2 |
| Lignin (g kg$^{-1}$) | 198 | 48 | 90 |
| Polyphenols (g kg$^{-1}$) | 7.8 | 11.3 | 19 |
| Lignin/N ratio | 6.9 | 6.2 | 2.6 |
| Kg N in 4.0 t C ha$^{-1}$, -N [+N] | 324 [564] | 68 [308] | 323 [563] |

155



### 2.1.3 Sample collection

The field campaign took place in February 2021, 19 years after the establishment of the long-term field trial and after 38 growing seasons of maize. Undisturbed soil samples were collected down to 70 cm using a gauge auger, one core was taken near the middle of each sampled plot. Subsections of 5 cm were cut, weighed and a 5 g subsample of each sample was immediately placed in an air-tight recipient to determine the water content and calculate the bulk density (see Sect. 2.2.1). The samples were sieved through a 8 mm sieve on the day of collection and air-dried in Kenya. Back in the lab, ca. one week later, the samples were sieved through a 2 mm sieve and the large organic residues (> 2 mm) that passed through the sieve were removed by hand. The samples were dried at 45°C. Using a soil splitter, samples were separated homogeneously in two parts, with one part being ground and the other part stored without grinding. The stocks of SOC were determined for the following depth layers: 0-5 cm, 5-10 cm, 15-20 cm, 25-30 cm, 35-40 cm, 45-50 cm, 55-60 cm, 65-70 cm. Each treatment (including the control) had two variations (with and without mineral nitrogen input), each variation was replicated 3 times (i.e., 4 treatments x 2 variations x 3 replicates = 24 plots), and 8 depth layers were selected, which leads to a total of 192 samples.

## 2.2 Laboratory analyses

### 2.2.1 Bulk density

Soil samples were collected using a gauge auger, reaching depths of up to 70 cm. Given the auger's 50 cm length, samples were obtained in two stages: the first sample down to 50 cm, followed by a second sample from 50 to 70 cm using an extension for the auger. Subsections of 5 cm were cut, weighed and a 5 g subsample of each sample was immediately placed in an air-tight recipient for measuring the water content. Samples were air-dried in Kenya and, upon arrival in Switzerland, water content was measured, and the dried weight of each subsection was calculated. The dried weight of each subsection and the inside volume of the auger were used to calculate the bulk density. Disturbance to the 45-50 cm and 50-55 cm sections during initial sampling necessitated the replacement of the 45-50 cm section's bulk density with that of the 40-45 cm section (the 50-55 cm section did not need to be adapted as it was not used for further analysis).

### 2.2.2 Soil organic carbon stocks and $\delta^{13}$C values

Before the analysis of organic carbon and nitrogen, large organic matter particles were removed with tweezers after the sample was spread on a smooth surface. Samples were not treated with HCl before the analysis of OC as the pH of all samples was ≤ 6. In addition, a selection of samples was tested with HCl (6M) and did show a reaction, confirming the absence of carbonates. Afterwards, 11 to 15 mg of dried and grinded soil was weighed in a tin cup. The samples were analyzed using an elemental analyser-isotope ratio mass spectrometer (Thermo Flash HT/EA or CE 1110 coupled to a Delta V Advantage), to obtain the organic carbon and nitrogen concentrations and $\delta^{13}$C values. The OC concentration was subsequently converted to stocks using the measured bulk density.

### 2.2.3 $\delta^{14}$C values



Homogenized samples were fumigated in silver capsules (Elementar) with HCl (37%) for 72h to remove carbonate (Komada

et al., 2008). Fumigated samples were then neutralized for 24 hr at 60°C over solid NaOH to remove residual acid. The samples were then wrapped in a $8 \times 8 \times 15$ mm tinfoil boat (Elementar) and pressed prior to analysis. Fumigated samples were analyzed for $\delta^{14}C$ using a Gas Ion Source of Micadas system at ETH Zurich. Samples were normalized using oxalic acid II (NIST SRM4990C). The Measured $\delta^{14}C$ / $\delta^{12}C$ ratios are reported as $F^{14}C$, as established by Stuiver and Polach in 1977 (Reimer et al., 2004).


### 2.2.4 Effective cation exchange capacity

Before the analysis of the effective cation exchange capacity (CEC), organic matter macro particles were removed from the soil sample. Subsequently, 2 g of dried and milled soil was measured and placed in a centrifugation tube, and 25 ml of 0.01M $BaCl_2$ was added. The sample was then agitated on a reciprocal shaker at 150 rpm for 2 hours. Next, the sample was

centrifugated for 10 minutes at 2500 rpm. The supernatant was then filtered through a Whatman 41 filter. After filtration, 1 ml of the solution was diluted in 4 ml of water (1/5 solution). The diluted solution was analyzed using ICP-OES (G8010A Agilent 5100 SVDV ICP-OES, Parent Asset SYS-10-5100). The concentrations of Al, Ca, K, Mg, Mn, and Na in the solution were subsequently converted to cmol per kg of soil before calculating the effective cation exchange capacity by summing the concentrations of all the aforementioned cations (Hendershot and Duquette, 1986). CECeff was measured for the 0-5 cm, 35-

40 cm, and 65-70 cm depth layers.

### 2.2.5 Soil texture

To analyze soil texture, 200 to 300 mg of 2 mm-sieved soil was weighed in a glass jar and treated 4 times with 10 ml $H_2O_2$ in

a warm water bath (ca. 50 °C) to completely remove organic matter. The samples were then transferred into a in a plastic tube and 7 ml of 10 % $(NaPO_3)_6$ was added and shaken overnight to completely disperse the soil minerals. The particle size was analyzed using a particle size analyzer (LS13320 Beckman coulter) to obtain the fraction clay (vol% < 2 μm), silt (vol% 2 – 53 μm) and sand (vol% > 53 μm). Texture was measured for the 0-5 cm, 45-50 cm, and 65-70 cm depth layers.

### 2.2.6 Soil pH

The soil pH was determined using a pH meter (Thermo Scientific™ Eutech™ "150 Series Waterproof Handheld Meters"). To prepare the soil samples, 10 g of dried soil was placed in a tube and 25 ml of deionized water was added. The tubes were shaken on a reciprocal shaker at 150 rpm for 2 hours. Afterward, the slurry was allowed to settle for 24 hours. Prior to measurement, the pH electrode was calibrated using buffer solutions with pH values of 7 and 4. Care was taken to ensure that

the diaphragm of the electrode did not come into contact with the soil particles. The pH was considered stable when the measured pH value remained constant within 0.02 units over a period of 5 seconds. The soil pH was measured for the 5-10 cm, 15-20 cm, 35-40 cm, and 55-60 cm depth layers.



**2.3 Statistical analysis**

**2.3.1 Linear mixed effect models of the treatments effect over the depth profile**

To test which measured variables are significantly affecting SOC stocks along the depth profile, a linear mixed effect model was constructed. Linear mixed effect models are linear models that allow to separate an independent variable into fixed effects and random effects. They are particularly useful when more than one measurement is made on a given statistical unit. It is therefore suitable for this work, since there was more than one sample for each site. The dataset was tested for outliers using

the boxplot method with the R function "identify_outliers", after which no outliers were identified (Kassambara, 2022). Normality of the data was confirmed visually (Fig. S1a), and homogeneity of variance was verified, showing no evident relationship between residuals and fitted values (Fig. S1b). The linear mixed effect model was built with the "lmer" function of the package "lmeTest" (Kuznetsova et al., 2017). The model included SOC as dependent variable, the site as a random effect and silt content, clay content, and the interaction between depth, organic residue input and mineral fertilizer (i.e., *+N* or

*-N*) as fixed effects. Other measured parameters such as pH and CECeff were not considered in this model, because they correlated with the application of all three types of organic residues (especially manure) (Fig. 1).

**2.3.2 Two sample t-tests between treatments**

To evaluate the impact of organic residue treatments on SOC stocks along the soil profile, we examined SOC stocks at different

depth intervals. This analysis involved conducting a sample t-test for each organic residue treatment to assess the difference in SOC stocks between the treatment and the control across various depth layers. The depth layers examined included the OC stocks of the entire profile (0-70 cm), of two cumulated depth layers (0-30 cm and 30-70 cm), and of each individual 5 cm depth layer (0-5 cm, 5-10 cm, 15-20 cm, 25-30 cm, 35-40 cm, 45-50 cm, 55-60 cm, and 65-70 cm). In order to increase the statistical power, statistical analyses were performed by combining the *+N* and *-N* sub-treatments of the same treatment,

focusing therefore only on the effect of organic residues application on SOC stocks. This approach was justified by the results of the linear mixed effect model, which indicated that mineral fertilizer was not significantly affecting SOC over the depth profile. As the sample size per depth layer was small, normality of the sample distribution was assessed with histograms, and there was no indication of significant departures from normality (Fig. S2). Homogeneity of the variance at each depth layer was assessed with the R function '*bartlett.test*' (R Core Team, 2022). Due to non-homogeneous variance of SOC stocks between

treatments across various depth layers, we conducted a Welch two-sample t-tests between the control and each organic residue treatment at every depth layer mentioned earlier. This analysis was performed using the "t.test" function in R. Treatments were considered significantly different than the control when their *p*-value was < 0.05. In the analysis of each 5 cm depth layer, if one layer showed a *p*-value greater than 0.05, while the two layers above and the two layers below showed significant differences compared to the control, the lack of significance in the layer above was attributed to sampling variability.



Consequently, the treatment was considered to affect OC stocks down to the deepest layer where a significant deviation from the control was observed. This was only the case for the *Manure* treatment at depth 25-30 cm.

The impact of organic residue treatments on soil pH and effective cation exchange capacity (CECeff) was further assessed using a Welch two-sample t-tests, similar to the approach used for the analysis of SOC stocks. These tests compared organic

residue treatments with the control treatment, with all treatments including combined *+N* and *-N* sub-treatments, at each individual depth layer where these parameters were measured.

To validate the findings of the linear mixed model regarding the effect of mineral fertilizer, we conducted further analyses by comparing the difference between the *+N* and *-N* versions of each organic residue treatment (as well as the control). This

comparison was performed using a Welch two-sample t-tests across the same depth layers used in the analysis of the effects of organic residues (i.e., 0-70 cm, 0-30 cm, 30-70 cm, 0-5 cm, 5-10 cm, 15-20 cm, 25-30 cm, 35-40 cm, 45-50 cm, 55-60 cm, and 65-70 cm). The results indicated no significant difference for any treatment at any depth layer. This supports the methodology of combining *+N* and *-N* treatments for the analysis of the effect of organic residues.

**2.3.3 Power analysis of statistics on 5 cm depth layers**

To determine whether the number of replicates was optimal for testing the difference between OR application and control, a power analysis was performed using the R function "pwr.t.test" (Champely, 2006). The analysis was performed twice and both times the Cohen's d effect size measure (i.e., standardized difference between two means) was used as an input, along with a significant level of $\alpha = 0.05$. The first analysis determined the actual power, based on the number of samples used in

each statistical analysis we performed. The second analysis was to calculate the number of samples needed to reach a power of 80% (i.e., i.e., the probability that the test correctly rejects the null hypothesis when the alternative hypothesis is true) and therefore power = 0.8 was used. This was performed for the analyses done at each individual 5 cm depth layer (0-5 cm, 5-10 cm, 15-20 cm, 25-30 cm, 35-40 cm, 45-50 cm, 55-60 cm, and 65-70 cm) to assess if our study design had sufficient statistical power to determine the exact depth layer down to which the ORs were significantly affecting OC stocks.

**3 Results**

**3.1 Soil characteristics**

The incorporation of organic residues, particularly manure, resulted in overall higher pH$_{H2O}$ values compared to the other treatments, and increased the effective cation exchange capacity (CECeff) (Fig. 1). Although the differences were not always significant between the treatments at the same depth, the tendency was consistent across all depth layers, suggesting that the

effect of organic matter application on soil properties may not be limited to the topsoil. Variability in soil pH$_{H2O}$ and CECeff between treatments was greater in the topsoil, ranging from 4.2 to 6.6 for pH and 5.3 to 19.0 (meq/100g) across all treatments.





Mineral N fertiliser application had a non-significant, but consistently negative effect on pH and CECeff when no organic residues were added (Fig. S3). Soil texture was uniform across all treatments with average clay, silt and sand proportions of 70%, 26.6% and 1.6% respectively (Fig. S4).


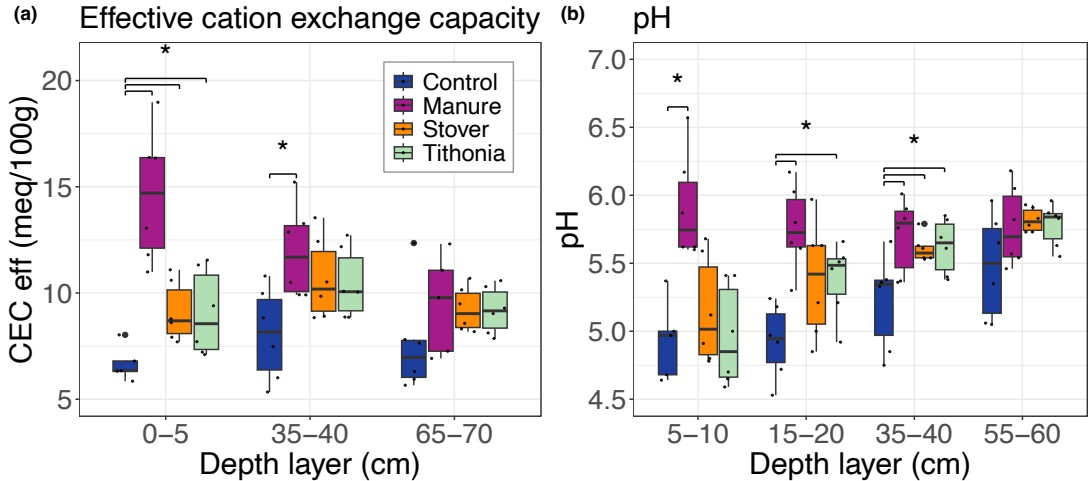

*Figure 1: Boxplots of (a) effective cation exchange capacity (CECeff) and (b) pH$_{H2O}$ for organic residue treatments at different depth layers. Significant difference between treatments at a similar depth are highlighted with a "*".*

## 3.2 Effect of organic residues on SOC stocks

**3.2.1 Effect of organic residue inputs on SOC stocks in the top- and subsoil**

When calculating the OC stocks for the topsoil and subsoil separately we found that, on average for all treatments, the subsoil layer (30-70 cm depth) contained 48.5% ± 1.7% of the total stocks of the entire 0-70 cm soil profile (Fig. 2). Considering the whole measured profile, every treatment receiving organic amendments (combined for +/- N) led to OC stocks that were significantly higher than OC stocks of the control treatment. It is noted that this difference was significant mainly because of

the high differences in the topsoil layer. Only the *Manure* treatment was significantly different from *Control* in the subsoil (i.e., the 30-70 cm depth layer)





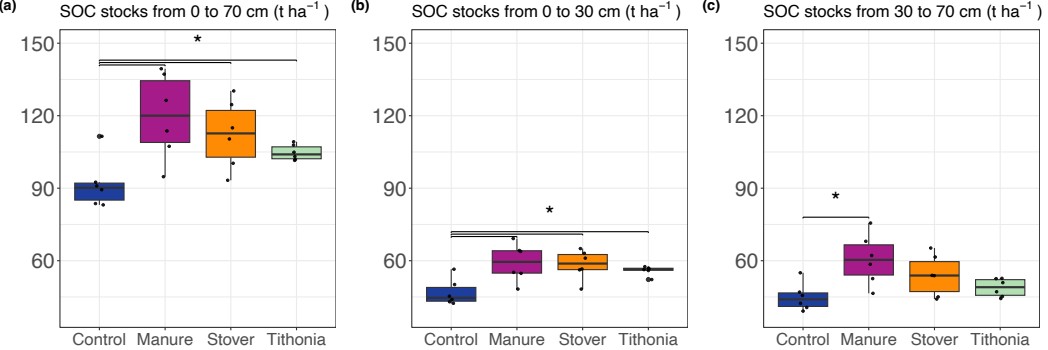

*Figure 2: Total stocks of organic carbon for (a) the 0-70 cm soil layer, (b) the topsoil (0-30 cm) and (c) the subsoil (30-70 cm). Significant difference between organic residues treatments and control are highlighted with a "*".*

### 3.2.2 Effect of organic residue inputs on SOC stocks along the soil profile

The difference in SOC stocks between the organic residue treatments and the control was most pronounced in the topsoil and gradually decreased with depth (Fig. 3). The linear mixed effect model revealed depth as the dominant factor influencing SOC stocks when considering all measured depth layers (Table 2). Additionally, manure and stover inputs had statistically significant effects on SOC stocks, while the application of *Tithonia* residues and mineral N fertilizer did not. Texture did not significantly influence the SOC stocks.

Table 2: Summary statistics of the linear mixed effect model for soil organic carbon stocks on all measured depth layers

|  | estimate | Standard error | *p* value | |
| --- | --- | --- | --- | --- |
| **(Intercept)** | 8.49 | 0.70 | 1.52E-14 | * |
| **Depth** | -0.06 | 0.01 | 4.82E-09 | * |
| **Manure** | 2.18 | 0.89 | 0.02 | * |
| **Stover** | 2.69 | 0.89 | 0.01 | * |
| Tithonia | 1.60 | 0.89 | 0.08 | |
| Mineral N | -0.10 | 1.02 | 0.92 | |
| Silt | -0.003 | 0.01 | 0.77 | |
| Sand | 0.01 | 0.03 | 0.79 | |
| Depth:Manure | 0.00 | 0.01 | 0.75 | |
| **Depth:Stover** | -0.03 | 0.01 | 0.03 | * |
| Depth:Tithonia | -0.01 | 0.01 | 0.45 | |
| Depth:Mineral N | 0.03 | 0.02 | 0.12 | |
| Manure:MineralN | 0.92 | 1.34 | 0.50 | |
| Stover:Mineral N | -0.39 | 1.34 | 0.77 | |
| Tithonia:Mineral N | 0.54 | 1.42 | 0.71 | |
| Depth:Manure:Mineral N | -0.03 | 0.02 | 0.14 | |
| Depth:Stover:Mineral N | 0.00 | 0.02 | 0.94 | |
| Depth:Tithonia:Mineral N | -0.04 | 0.02 | 0.09 | |





Across individual measured 5 cm layers in the topsoil (i.e., within the 0-30 cm depth layer), OC stocks ranged from 5.5 to 13.1 t OC ha⁻¹ per 5 cm depth layer (Fig. 3). The effect of organic residues on OC stocks per 5 cm layer revealed down to which depth the different organic residues had a significant effect on SOC stocks (Fig. 3). In the top 0-5 cm, *Manure* had the highest OC stocks with 11.2 ± 1.6 t ha⁻¹ and *Control* had a significantly lower OC stock with 8.3 ± 0.4 t ha⁻¹. *Manure* was affecting OC stocks the deepest, with a significant effect down to 60 cm. At this depth layer (i.e., 55-60 cm), *Manure* had an OC stock

of 7.2 ± 1.8 t ha⁻¹ and *Control* of 5.2 ± 0.6 t ha⁻¹. *Stover* was significantly different from *Control* down to 40 cm. The OC stocks of the *Tithonia* treatment were only significantly different from *Control* in the top 5-20 cm.

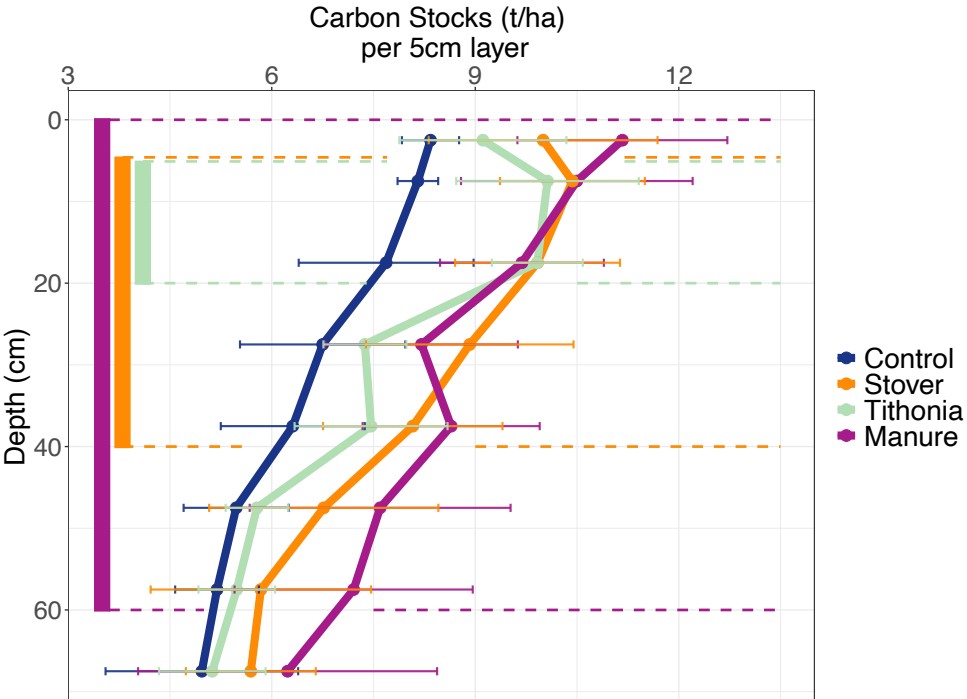

*Figure 3: Profile of SOC stocks for each measured 5 cm layer (not cumulative). The vertical bars on the left side of the graph show the depth layers over which the SOC stocks for the different treatments are significantly different from the control treatment.*

The power analysis indicated a high risk of Type II errors (i.e., incorrectly rejection that there is a significant difference),

particularly in soil layers below 20 cm. For each OR treatment and at each depth layer, the power of the test fell below 80% when no significant difference was detected between the OR and the control (Fig. 4a). This suggests that for these statistical tests, there is a greater than 20% chance of failing to detect a true effect. This implies that the absence of a significant difference





between *Control* and either the *Stover* or *Tithonia* treatments in deeper soil layers should be interpreted with care. This is possibly due to the increased data variability with increasing depth. The deeper down the soil profile, the more samples are

thus needed to detect a true effect of the treatment (Fig. 4b).

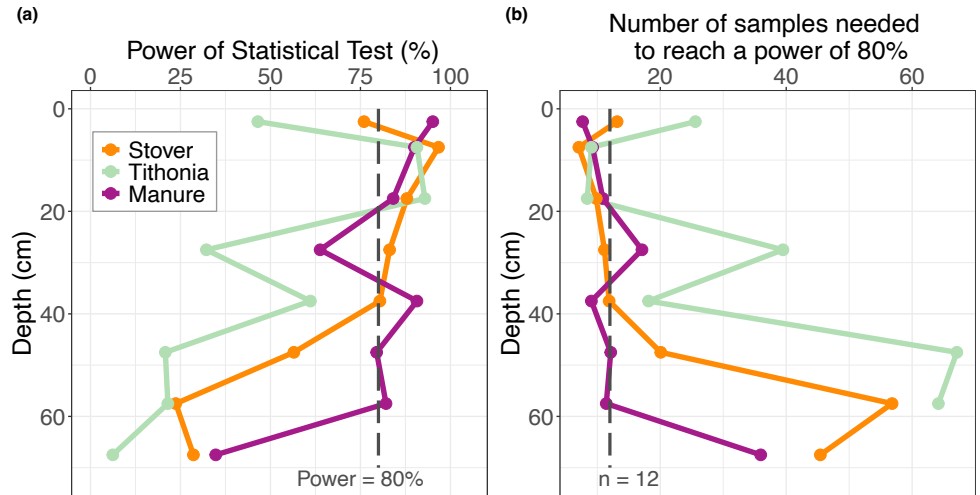

*Figure 4: (a) Power of statistical analysis at each depth layer and for each organic residue treatment and at each 5 cm depth layer. The 80% power threshold is indicated by the dashed line. (b) The amount of sample samples required to reach a*

*statistical power of 80% for each organic residue treatment and each 5 cm depth layer. The actual number of sample (n = 12) is indicated by the dashed line.*

### 3.2.3 Effect of treatments on SOC stocks considering +*N* and -*N* treatments separately

The SOC stock in the 0-30 cm layer was highest for *Manure-N* and lowest for *Control+N* with values of $60.9 \pm 8.0$ t OC ha$^{-1}$ and $44.1 \pm 2.6$ t OC ha$^{-1}$, respectively (Fig. 5a). Similarly, over the whole measured profile (0-70 cm depth), *Manure-N* showed the highest OC stocks, while *Control+N* showed the lowest, with mean values of $124.4 \pm 24.5$ t OC ha$^{-1}$ and $85.9 \pm 15.7$ t OC ha$^{-1}$ respectively (Fig. 5a). This means that after 38 growing seasons over 19 years, and considering the top 70 cm of the soil, *Manure-N*, as the most effective treatment to limit SOC losses, could maintain $38.5 \pm 8.8$ t OC ha$^{-1}$ more SOC compared to

the least efficient treatment, *Control+N*. Adding mineral nitrogen resulted in consistently lower OC stock across the soil profile for *Control*, but not for *Manure, Stover* and *Tithonia* (Fig. 5b). However, the difference between +*N* and -*N* variation of each treatment was never statistically significant for any depth layer.



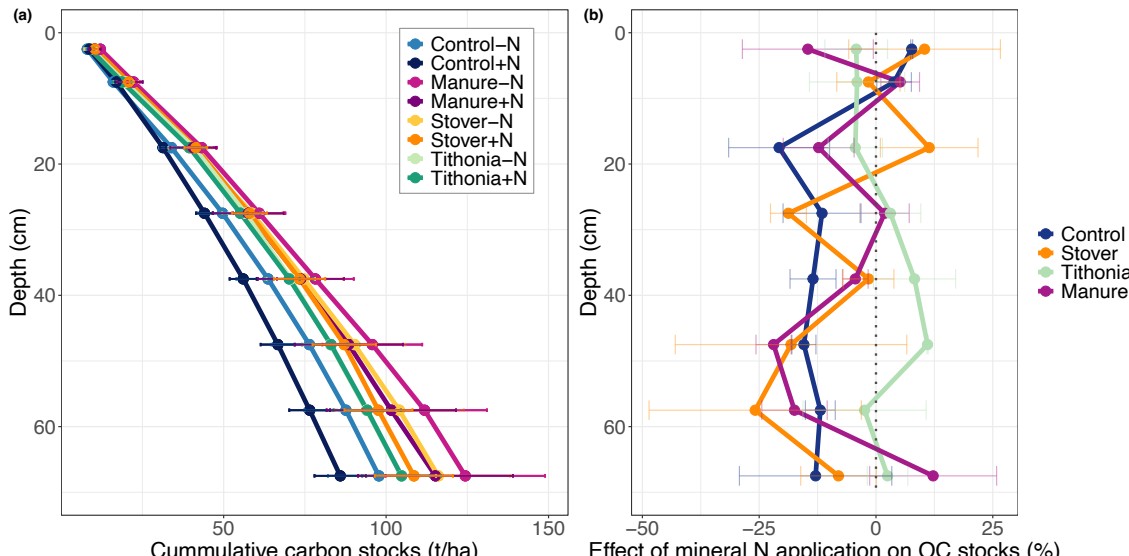

Figure 5: (a) Cumulative stocks of organic carbon along the depth profile including -N and +N treatments and (b) difference in OC stocks between +N and -N treatment at each measured 5 cm depth layer (not cummulative); values below 0 indicate a loss in OC when mineral nitrogen is applied. The difference SOC stocks between -N and +N treatments were never significant.

## 3.3 C/N ratio and stable isotopes of organic carbon

The C/N ratio of organic matter had values between 10 and 12 along the soil profiles and decreased with depth below 30 cm (Fig. 6a). The C/N ratio across the depth profiles was not significantly different between treatments, but below 30 cm, it tended to be lower in the *Control* treatment compared to all organic residue treatments. This difference was primarily due to the consistently lower C/N ratio of the *Control+N* treatment at intermediate depths (Fig. S5).

Depth profiles of the $\delta^{13}C$ value of SOC show that the largest differences between the treatments are in the top 20 cm of the soil (Fig. 6b). The lowest values are for the *Manure* treatment (-19.5 – -20 ‰), while the *Stover* and *Control* treatments have the highest values (ca. -18 ‰). The $\delta^{13}C$ value for the *Tithonia* treatment is intermediate (ca. -18.5 ‰). Below this depth, all treatments have similar $\delta^{13}C$ values in the range of -17.5 – -18.5 ‰, with a tendency for lower values in the *Manure* treatment. The $\delta^{13}C$ value of SOC of the *Stover* treatment is within the range of *Control* all along the profile. Additionally, the $\delta^{13}C$ signatures of all treatments at depths below 50 cm are similar, suggesting that this is the $\delta^{13}C$ signature of the soil at that depth before the experiment started.



As observed in the depth profile of radiocarbon (F$^{14}$C), the OC becomes older with depth, i.e., the lower the F$^{14}$C value (Fig.
6c). However, below the surface layer (i.e., the 0-5 cm depth layer) the F$^{14}$C value are higher (i.e., the OC is younger) for all
three treatments receiving organic amendments, compared to the control treatment, suggesting that organic residue input
affected OC below the layers where it is applied. The higher F$^{14}$C values for the *Manure* treatment in the deepest layer suggest
that this treatment had the largest impact on subsoil OC.

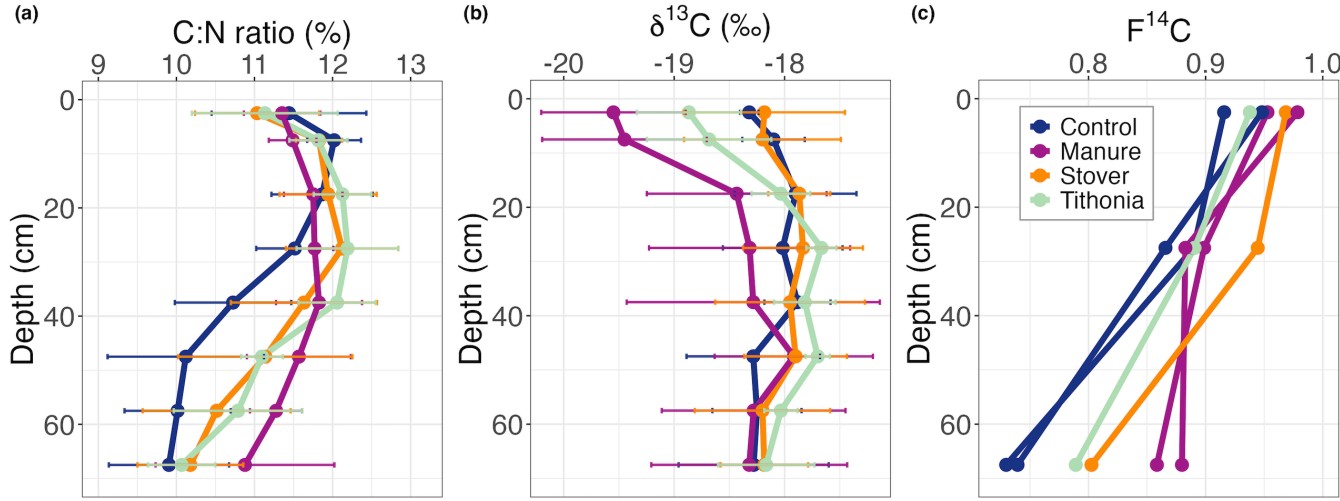


*Figure 6: Soil profiles of (a) C/N ratio, (b) δ$^{13}$C and (c) F$^{14}$C of the different organic residue treatments (for F$^{14}$C, only 6
plots were measured and there were two replicates for Control and for Manure)*

## 4 Discussion

Our results indicate that about 50% of the SOC stock measured down to 70 cm is located below 30 cm, showing the importance
of subsoil OC stocks. Moreover, the addition of manure, *Tithonia* residues and maize stover all had a positive impact on SOC
stocks, limiting the loss of organic matter compared to the control treatment. Although this effect was greatest in the top 20
cm, it could be observed down to 40 cm and 60 cm for the Stover and the Manure treatments, respectively. The *Manure*
treatment led to the highest OC stocks down to 70 cm. However, a power analysis indicated that more samples are required to
accurately assess whether the effect of organic residues on OC stocks in the subsoil is significant or not.


### 4.1 Application of manure affects OC the subsoil

Our study shows that the application of manure affects SOC stocks in the subsoil in a tropical long-term field trial. Among all
organic residues applied, only *Manure* had OC stocks that were significantly larger than *Control* in the 30-70 cm depth layer
(Fig. 2), while the application of manure was significantly impacting OC stocks down to 60 cm (Fig. 3). This shows that
organic residues can also influence SOC in the subsoil. In addition, the Δ$^{14}$C values along the soil profile suggest that the



addition of Manure led to the addition of fresh OC down to a depth of 65-70 cm. However, this effect wasn't mirrored in the $\delta^{13}$C values at the deeper layer, suggesting that only a small portion of the added organic amendments made their way downwards.

Our findings are in line with Leuther et al. (2022), who found that high levels of manure application in a long-term (36 years) field study in Germany increased OC stocks at 35-45 cm depth, and by Anandakumar et al. (2022), who observed a significant increase in OC stock between 25 and 50 cm depth after applying manure in a semi-arid tropical Alfisol in India. Also, a study from Sommer et al. (2018) revealed that manure was able to slow steady losses of SOC from topsoils of Kenyan Ferralsols over time, but only when at least 4.0 t C ha$^{-1}$ yr$^{-1}$ was applied. While this is the same amount as what was applied in our study,

it is the largest amount compared to common range of 0.9 to 4 t C ha$^{-1}$ yr$^{-1}$ that are applied by Kenyan farmers who own cattle (Tittonell et al., 2008). Therefore, the impact of using manure may not be as great in the deeper layers when considering a scenario outside the context of long-term field trials. For instance, Laub et al. (2023) did not observe a significant effect on topsoil OC when the annual application of manure was only 1.2 t C ha$^{-1}$ yr$^{-1}$.

**4.2 The effect of *Tithonia diversifolia* and maize stover residues on OC could only be detected in the topsoil**

In contrast to the *Manure* treatment, our results showed that *Tithonia* and *Stover* did not affect the subsoil OC stocks significantly in the 30-70 cm layer (Fig. 2). Also, no significant differences below 20 cm and 40 cm for *Tithonia* and *Stover*, respectively, were observed in the individual 5 cm depth layers (Fig. 3). However, for both treatments, the statistical power was below 80% at every depth layer where the treatments were not significantly different from the control. This suggests that

our sample size was insufficient to prevent Type II errors in the subsoil (Fig. 4a and 4b). One must therefore be cautious when interpreting these non-significant differences. Moreover, both *Tithonia* and *Stover* treatments showed signs of younger OC below 30 cm depth compared to the *Control*, as indicated by the higher F$^{14}$C values (Fig. 6c). Nevertheless, these effects in the 65-70 cm depth layer were less pronounced compared to *Manure*. This suggest that the effect of stover and tithonia residues can be traced below 30 cm, but that this effect is not strong enough to detect a significant impact on subsoil OC stocks. More

samples would be needed to conclusively reject the hypothesis that application of *Tithonia* and *Stover* lead to higher OC stocks in the subsoil. Our observations differ from observations from Córdova et al. (2018), who found that high-quality residues led to a higher accumulation of carbon in the soil compared to low-quality residues, because *Stover* had higher OC stocks than *Tithonia* in the topsoil and in the subsoil. Nevertheless, our results are in line with the meta-analysis results of Gross and Glaser (2021) who found manure to be more effective in stabilizing SOC than green manure. Their meta-analysis was

dominated by temperate soils, but here we can confirm their results for tropical soils.

The $\delta^{13}$C signatures of OC in the topsoil under the *Stover* and *Control* treatments were similar. However, as maize is a C4 plant, it would be expected that if more stover residues was incorporated in the soil compared to the control treatment, the $\delta^{13}$C signature of SOC would be highest under the stover treatment (Tieszen, 1991). As the *Stover* treatment received 4 t C ha$^{-1}$ yr





[1] more maize derived residues (i.e., stover) compared to the *Control* treatment, the similar $\delta^{13}$C value in the topsoil shows that the additional organic matter inputs for the *Stover* treatment did not results in additional SOC sequestration compared to the control treatment. However, as topsoil OC stocks were significantly larger for the *Stover* compared to the *Control* treatment, this shows that the larger SOC stocks were due to lower losses of initial SOC in the *Stover* treatment, and not to additional SOC sequestration.


Concerning the *Tithonia* treatment, the lower $\delta^{13}$C values of topsoil OC and the significantly larger topsoil SOC stocks compared to the *Control* treatment, suggest that the latter is at least partly caused by the sequestration of OC originating from the *Tithonia* residues. The limited effect of both the *Stover* and *Tithonia* treatments on topsoil $\delta^{13}$C values suggests that both residues are being quickly mineralised upon application, which explains why they do not compensate the steady loss of OC in

both treatments.

### 4.3 Mineral nitrogen does not significantly affect SOC stocks

The linear mixed effect model did not select mineral N as a parameter having a significant impact on the SOC stocks. This corroborates the meta-analysis conducted by Gram et al. (2020) on agricultural practices in sub-Saharan Africa, which showed that mineral N fertilizer had no significant impact on topsoil SOC concentrations. However, the interaction between SOC and

mineral fertilizer is complex and two global meta-analyses on the effect of mineral N on agricultural topsoil, conducted by Ladha et al. (2011) and Liu et al. (2023), suggested that the addition of mineral N fertilizer typically has a positive impact on SOC content. One reason for this impact is that as mineral fertilizer tends to enhance biomass production, it commonly leads to an increase in the quantity of plant residues returned to the soil, thereby positively impacting the soil organic matter content (Geisseler and Scow, 2014). This does not apply at our study site in Embu, as the yield was not much responsive to N

fertilization and crop residues were removed from the field after harvest (Laub et al., 2023a). Also, while Liu et al. (2023) emphasize the significance of specific climatic and environmental factors on the effects of mineral fertilizer application on SOC, neither of the two meta-analyses included studies from eastern Africa (Ladha et al., 2011b; Liu et al., 2023). For instance, in a study conducted in Kenya, Ndung'u et al. (2021) observed a decrease in topsoil OC concentration after the application of mineral nitrogen to Ferralsols, while Laub et al. (2023b) did not observe a consistent effect of mineral N on SOC stocks in

four long-term agricultural field trials in central and western Kenya.

The *Control* treatment had the largest difference between the *+N* and the *-N* treatments when considering OC stock over the whole 0-70 cm depth profile, i.e. *Control+N* had a consistently lower OC stock than *Control-N* below 10 cm (Fig. 5a). While this difference was not significant, this suggests that the sole application of mineral nitrogen increased the rate of OC losses at

the study site. One possible explanation for this observation is that the application of mineral fertilizer stimulates the activity of fast-growing microorganisms that can lead to a priming effect, accelerating the decomposition of organic matter in the soil (Chen et al., 2014). It is also possible that mineral nitrogen alone leads to increased rates of OC loss because it increases the



degradation of below-ground plant residues; for instance mineral fertilizer was shown to increase the degradation of young C4 lignin (Hofmann et al., 2009).

### 4.4 The potential of organic inputs to reduce SOC losses in tropical agroecosystems

There is a motivation to adopt agricultural practices to reduce the atmospheric $CO_2$ concentration by increasing SOC storage (Bossio et al., 2020; Minasny et al., 2017). According to Corbeels et al. (2019), there is a potential to achieve this in Sub-Saharan Africa, not primarily as a measure to mitigate climate change but to improve crop productivity. However, the low $F^{14}C$ values measured in each treatment, especially in the subsoil, indicate that very little of the freshly added organic matter remains in the soil, suggesting that even the application of significant amounts of OR would not contribute substantially to climate change mitigation. This aligns with the findings of Sierra et al. (2024) who demonstrated that only a minimal amount of new carbon inputs make their way through the profile and have the potential to remain stable for periods extending beyond 50 years. Also, results from Laub et al. (2023b) show that all treatments of the long-term trial in Embu have lost topsoil OC during the first ca. 20 years of the field trials, despite differences in magnitude between treatments. In accordance with the results from Laub et al. (2023b) for topsoils (0-15 cm), we show that after 19 years, OC stocks down to 70 cm was largest for *Manure-N* and lowest for *Control+N* with 124 t ha$^{-1}$ and 85 t ha$^{-1}$ respectively. This is a difference of 39 t C ha$^{-1}$, which represent about 50% higher OC stock in *Control+N* than in *Manure-N* stocks. Therefore, although there might be little potential to effectively reduce atmospheric $CO_2$ concentration by increasing SOC storage in soils with mainly low activity clays that were recently converted to agriculture, there is a potential to optimize agricultural nutrient management to minimize SOC losses after land use conversion from forest to agriculture and thereby forming a carbon sink compared to business as usual scenarios (i.e. low input of mineral and organic resources).

### 5 Conclusion

Our study shows that organic amendments have an influence on the SOC content throughout the soil profile in a tropical arable soil. Using soil samples down to 70 cm from a long-term field trial in central Kenya, we show that the application of 4 t C ha$^{-1}$ yr$^{-1}$ in the form of manure led to significantly larger SOC stocks down to 60 cm compared to a control treatment. In contrast, the application of residues of *Tithonia diversifolia* and maize stover impacted SOC stocks to a lower extent, while this effect was limited to the topsoil. However, due to insufficient statistical power for the subsoil for these treatments, further research with larger sample sizes is essential to draw definitive conclusions on their effects. The application of mineral nitrogen did not contribute positively to the effect of organic residues on SOC stocks, and even had a negative effect in the control treatment. Our results show that the application of manure is the most appropriate nutrient management strategy to limit losses of SOC in recently established croplands on the studied tropical soils. Although our study shows that the studied nutrient management strategies did not lead to a net increase in SOC stocks in the investigated soils, we argue that appropriate agricultural management would partly mitigate the effect of deforestation on the concentration of atmospheric $CO_2$, while simultaneously improving soil quality for food production.



**Code availability**

The code can be provided upon request.

**Data availability**

The data used in this study are open access and available at https://figshare.com/s/c2f2787b7a56ef7ad656.

**Authors Contribution**

Conception and design of the field campaign: MVdB and JS.
Design of the experiment: MVdB and JS.
Performing the experiments: CRM
Analysis and interpretation of the data: CRM, MVdB.
Writing the manuscript: CRM and MVdB with inputs from all co-authors.

**Competing interests**

The contact author has declared that none of the authors has any competing interests.

**Acknowledgements**

We thank Britta Jahn-Humphrey for help with soil processing, and ChatGPT, an AI language model developed by OpenAI, for assistance in the writing process.

**Financial support**

This work was supported by the Swiss National Science Foundation SNSF (Ambizione project number PZ00P2_193617 granted to MVdB).

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
