# Peer review of "Depth Effects of Long-term Organic Residue Application on Soil Organic Carbon Stocks in Central Kenya"

_EGUsphere, 2024_

## Author Comment (AC1)

Dear Referees,

Thank you very much for your valuable feedback, suggestions and inputs. This will undoubtedly contribute to the enhancement of manuscript's quality. Below you will find a general outlook describing how we plan to address the comments that you provided.

Introduction, study design and discussion
- The flow between the paragraphs of the introduction will be improved. The C isotopes will be better introduced and research questions will be merged into one clear question: "Down to which depth do the different organic residues affect SOC stocks".
- The subsoil will be more clearly defined in the introduction, with an explanation for the 30 cm theshold between top- and subsoil.
- The specific recommendations to change phrasing will be considered and the text will be improved accordingly.
- Overall, some more detail will be added on the pH, CEC and C isotopes and on how these parameters are affected by organic residues in the introduction and in the discussion.
- Our study shows that organic residues affect SOC below the commonly studied soil depth of 30 cm. However, it remains unclear whether increased subsoil OC can impact crop productivity. This aspect will be emphasized more, as it warrants further investigation.
- It will be clarified that the results from Laub et al. (2023b) were based on a different sampling campaign than the samples used in our study.
- In Section 2.1.1, we will add a statement clarifying that "*all results indicating higher SOC for a given treatment in our study should be interpreted as losing less carbon, as Laub et al. (2023b) indicated that all treatments have been consistently losing SOC since the initiation of the experiment.*".
- The explanation of OC stocks calculations will be clarified.
- The limitations of our results will be further highlighted, and their interpretation will be presented accordingly. The reasons behind the choices of the applied statistical analyses will be provided. For example, we will put more emphasis on the fact that merging the ±N variant of the organic residues treatments has its limitation as the long-term field trial was designed to study the different impact of these treatments. However, we will highlight the fact that without merging the ±N variant the statistical power (which was already low for most analyses) would be much lower and inflate the risk of type II error.

Improvement of data anlysis:
- Following the feedback of the reviewers, the linear mixed model will be adjusted as follow:
  To avoid overfitting and ensure the robustness of our findings, we will exclude interactions in the model. This approach will simplify the model, focusing on the main effects and accounting for variability through random effects, thereby reducing the risk of capturing noise rather than true underlying patterns. We will evaluate the robustness of our model by fitting six different linear mixed-effects models using the lmer function from the lme4 package in R. Each model will include OC stocks as the response variable, the sampling block as a random effect, and different combinations of fixed effect predictor variables:
    • model 1 (depth, organic residues, mineral fertilizer, silt content, silt content)
    • model 2 (depth, silt content, silt content)
    • model 3 (depth, organic residues)
    • model 4 (depth, mineral fertilizer)
    • model 5 (organic fertilizer)
    • model 6 (mineral fertilizer)
  We will then compare these models with an ANOVA to determine the best-fitting model based on AIC, BIC, log-likelihood, and deviance values. Only the reuslts of the best fitting model will

be presented in the manuscript and additional information will be available in the suppementary information. To further test the effect of the organic residue treatment in the deeper soil layer, the same analysis will be performed on a subset containing only subsoil layers.

- We will justify the use of a *t*-test over other statistical tests and explain why we are convinced that the risk of a type I error is small compared to the risk of Type II error. We acknowledge the concern of reviewer 2 regarding type I errors with multiple *t*-tests. However, our study is hypothesis-driven, and we observe a clear trend: significant differences observed in the top layer diminish with depth. As we analyze deeper soil layers, the observed differences between OR treatments and the control decrease. We perform this analysis layer by layer to determine down to which depth these statistical differences are observed. We are not randomly testing a large number of treatmnents against each other (in which case the multiple *t*-tests would indeed increase the risk of type I errors). Given the limited data (due to sampling in a long-term field trial) and increasing variability with depth, we chose *t*-tests over an ANOVA because an ANOVA would be even more affected by the limited statistical power. When statistical significance was not achieved, the power was below 80%, supporting our choice. Therefore, we argue that the typical type I error inflation does not apply in our context and that we are more likely to experience type II error.

---

## Author Response (AR1)

Dear Referees,

Thank you very much for your valuable feedback, suggestions and inputs. This will undoubtedly contribute to the enhancement of manuscript's quality.
Responses from the co-autors are in green

**Reviewer 1**

This paper represents a nice study of the impacts of different forms of agronomic management (w/ and w/o N amendment & 3 different organic residues) on deep carbon stocks. As the paper highlights, this is an area that is in need of more study because most studies focus on the top layers of soil, whereas we know that much carbon, and especially stable carbon is in deeper soils. My biggest comment is on the statistical analysis, and potential interpretations. I applaud the authors for conducting a power analysis and being quite clear about where there are limitations on sample numbers. That said, I think there may need to be a little more caveating of the results earlier on in the study, and also an indication that there is also the potential for type I errors as well (I believe...). I'm also slightly concerned about model over-fitting with the inclusion of all of the interactions, given the number of samples present in the study. Especially given that almost none are significant, and also I don't think there is sufficient justification for inclusion of all of the interactions- ideally, the linear model should include interactions with sound theoretical justification and hypotheses around why these interactions might reasonable shift overall carbon levels, and specifically deep carbon.

An additional analysis that only subsets deep carbon may also be interesting to show, given that this is the piece that is particularly novel for this study. I presume that much of the presented model is driven by the carbon in the top soil. Some level of sensitivity analysis would also help make me more confident in the results of your model-e.g. do model results stand when you include fewer variables/interactions. This could be presented in supplements.

All these remarks were addressed and the manuscript was changed as follows:

Method
To avoid overfitting and ensure the robustness of our findings, we excluded interactions. This approach simplifies the model, focusing on the main effects and accounting for variability through random effects, thereby reducing the risk of capturing noise rather than true underlying patterns. We evaluated the robustness of our model by fitting six different linear mixed-effects models using the lmer function from the lme4 package in R. Each model included OC stocks as the response variable, the sampling plot as random effect and different combinations of fixed effect predictor variables: model 1 (depth, organic residues, mineral fertilizer, silt content, silt content), model 2 (depth, silt content, silt content), model 3 (depth, organic residues), model 4 (depth, mineral fertilizer), model 5 (organic fertilizer), model 6 (mineral fertilizer). We then compared these models with an ANOVA to determine the best-fitting model based on AIC, BIC, log-likelihood, and deviance values. To further test the effect of organic residue treatment application in the deeper soil layer, the same analysis was performed on a subset containing only subsoil layers. The R-squared, fixed effect

Results:
According to the ANOVA performed on all the different models, the model including only depth and the OR had the lowest AIC (596.05) and BIC (618.56), the highest log-likelihood (-291.03) and the lowest deviance (582.05), suggesting it was the best-fitting model among the four (Supplementaty table 1). The $R^2$ of this model was 0.76, representing the variation in carbon levels that this model explained, with 0.65 attributed to fixed effects and 0.11 to random effects. According to the model's result, depth, application of manure and application of maize stover were significantly impacting OC stocks, not the application of tithonia (Table 2). The same analysis performed on a subset conatining only subsoil data yileded very similar results with the best fitting model including only depth and organic residues as fixed effect and having a $R^2$ of 0.64. According to the subsoil best fit model, beside depth, only manure was significantly affecting OC stocks (Supplementary table 2). The application of mineral fertilizer was never considered to significantly affect OC stocks by any model.

Table 2: Summary statistics of the linear mixed effect model for soil organic carbon stocks on all measured depth layers

| | estimate | Standard error | p value | |
|---|---|---|---|---|
| (Intercept) | 8.98 | 0.41 | < 2e-16 | * |
| Depth | -0.07 | 0.00 | < 2e-16 | * |
| Manure | 2.04 | 0.53 | 0.001 | * |
| Stover | 1.59 | 0.53 | 0.007 | * |
| Tithonia | 0.92 | 0.53 | 0.095 | |

Supplementary table 1: ANOVA Comparison of Linear Mixed-Effects Models performed on all meaured depth layers

| Model | npar | AIC | BIC | logLik | deviance | Chisq | Df | Pr(>Chisq) |
|---|---|---|---|---|---|---|---|---|
| C ~ Mineral Fertilizer + (1 \| Plot) | 4 | 802.2 | 815.06 | -397.1 | 794.2 | | | |
| C ~ Depth + Mineral Fertilizer + (1 \| Plot) | 5 | 605.18 | 621.25 | -297.59 | 595.18 | 199.02 | 1 | <2e-16 *** |
| C ~ Depth + silt + sand + (1 \| Plot) | 6 | 608.27 | 627.56 | -298.13 | 596.27 | 0 | 1 | 1 |
| C ~ Organic Residue + (1 \| Plot) | 6 | 793.07 | 812.36 | -390.54 | 781.07 | 0 | 0 | |
| C ~ Depth + Organic Residue + (1 \| Plot) | 7 | 596.05 | 618.56 | -291.03 | 582.05 | 199.02 | 1 | <2e-16 *** |
| C ~ Depth + Organic Residue + silt + sand + (1 \| Plot) | 10 | 599.69 | 631.84 | -289.85 | 579.69 | 2.36 | 3 | 0.5013 |

Supplementary table 2: Summary statistics of the linear mixed effect model for soil organic carbon stocks on subsoil depth layers

| | estimate | Standard error | p value | |
|---|---|---|---|---|
| (Intercept) | 9.17 | 0.67 | < 2e-16 | * |
| Depth | -0.07 | 0.01 | 5.36e-10 | * |
| Manure | 1.86 | 0.61 | 0.007 | * |
| Stover | 1.03 | 0.61 | 0.11 | |
| Tithonia | 0.40 | 0.61 | 0.52 | |

Finally, I think this study should be published, but the language around your findings should be slightly less 'confident' given the results of your statistical analysis- this could just be done earlier on.
This was adapted in the new version of the manuscript

I'm a bit confused on the calculation of stocks given that I think you only measured a subsets of the soils at depth - in line 165 you say that you look at layers of 5cm increments, but these do not add up to the whole soil profile- do you assume that 45-50 is representative of 40-50? Perhaps I missed this explanation, but I think there should be some statement that makes clear what assumptions you are making on the calculation of stocks.
This was indeed misleading in the text and actually incorrect. What was actually done was that the bulk density values of the 45-50 cm and 50-55 cm layers were both replaced by the mean value of the 40-45 cm and 55-60 cm layers from the same profile. This was changed in the new version of the manuscript.

The method section o SOC stocks calculation was adapted as folows:
While bulk density was available for each depth layers in 5cm increments, the OC concentration was only measured on some layers (i.e., 0-5 cm, 5-10 cm, 15-20 cm, 25-30 cm, 35-40 cm, 45-50 cm, 55-60 cm, 65-70 cm). cm). For calculating the stocks over the whole profile, depth layers without SOC measurements were interpolated using available data from adjacent layers within each profile. This interpolation was performed using the approxfun function in R.

The intro covers some good ground but in its current form read a bit choppy. The isotope paragraph feels like its just floating and is not needed in the intro. The flow between paragraphs could also use a bit of work. A bit more context for the specific system could be useful in the final paragraph with the research questions. I also think that these questions still read somewhat broad, and contextualizing them to your system can add some useful

specificity.I personally also think its nice, even if briefly, to state what you expected out of these different schemes given the current literature.

The flow and introducton of the isotopes were improved in the new version of the manuscript.
The research questions were merged into one clear question: "Down to which depth do the different organic residues affect SOC stocks"

I'm not totally convinced that the isotope work fits in currently in a way that feels meaningful. The final data is somewhat mixed in its results and while I think its interesting, I'm not sure that in its current form that it is adding to the manuscript or primary findings. I think the paragraph in the intro feels quite plopped in, and while I'm not against its inclusion, the authors should work to make its inclusion feel less like an add-on.

The following are more specific/smaller comments that I think would strengthen the manuscript:

From this part onward, modification in the text are highlighted in yellow

33: citation for ...groundwater and the atmosphere; "it" should be clarified - I presume it is something like "intensive agriculture" or something?
In the long term, intensifying agriculture also leads to soil quality degradation (e.g., through crusting, compaction, erosion, and depletion of organic matter) (Lal, 2008).

41: OC is not defined in its first use.
Thus, maintaining SOC levels over time is crucial for sustaining soil productivity, especially in the highly weathered soils of tropical regions, which are particularly prone to Organic Carbon(OC) loss (Feller and Beare, 1997).

42: "This is clear from the fact" is unnecessarily wordy
Thus, maintaining SOC levels over time is crucial for sustaining soil productivity, especially in the highly weathered soils of tropical regions, which are particularly prone to OC loss (Feller and Beare, 1997). The conversion of tropical forests to agriculture typically results in an average reduction of organic carbon (OC) concentration in the topsoil (down to a depth of 30 cm) of around 50% within 25 years (Veldkamp et al., 2020).

49: Are there key functions other than nutrient management that you think are relevant to bring up? What are the key functions
Therefore, agricultural management should not only aim at increasing crop yield in the short term, but also at maintaining SOC at a level that maintains the key function that it regulates (e.g., nutrient cycling, water retention, improved cation exchange capacity (CEC), soil structure stabilization, promotion of biodiversity) in the long term.

66-67: "However" not "Nevertheless" works better; I don't think you have defined OR- I presume organic residue.
However, an incubation experiment on tropical soil indicated that, while this might be true in the short term, initial organic residues (OR) quality does not affect SOC accumulation in the long term (Gentile et al., 2011).

92: Not sure about this paragraph and how it fits in.
The isotope paragraph was changed and integrated differently

115: I'm not experienced in this region, but I'm a bit confused about the rain seasons- is it raning all year round? the given months span the entire year.
It is characterized by a continuous maize monoculture with two annual growing seasons that coincide with the bimodal precipitation pattern. The long rainy season generally occurs from March to September, gradually decreasing towards the end, leading into a dry period. The short rainy season spans from October to February, also tapering off into a dry period.

117: passive voice not needed "having been" => " the site was originally covered... before conversion to" would be clearer.
The site was originally covered by tropical evergreen forest before conversion to a low-intensity agricultural area.

125: Would be good to clarify that these treatments are the same as in this paper- this whole paragraph maybe feels like it should be in the intro justifying the gap in the literature of deeper depth investigations. This paper seems like a fundamental work that you build off of.

In a recent study conducted at this site, Laub et al. (2023b) showed that all nutrient management treatments at Embu resulted in significant SOC losses, but the use of farmyard manure was the most effective strategy to minimize the loss of SOC. Laub et al. (2023b) studied the Embu trial along with three other trials in different regions of Kenya. In their study on the Embu site, they observed consistent decreases in soil organic carbon (SOC) contents over time for all treatments (Fig. S8). The control treatment showed the greatest declines, with a loss of about 0.6-0.7 g C kg$^{-1}$ yr$^{-1}$. Despite differences in treatment effectiveness in maintaining OC, farmyard manure (4 t C ha$^{-1}$) emerged as the most effective strategy to limit SOC losses at Embu, with the lowest OC loss (i.e., 0.4 g C kg$^{-1}$ yr$^{-1}$). However, their focus was on the effects of treatments above 15 cm depths. These treatments and the overall experimental design (i.e., the Embu long term field trial) are the same as those investigated in this paper, which builds on their fundamental work by exploring the effects on deeper soil depths at one of the four long-term field trial sites.

139: Are these typical application rates for the region? Also some justification of Thitonia, corn stover, and maize (which I presume has happened in previous papers) would be good to include in brief somewhere. Perhaps in Study Area. Jumping back to "The organic resources" is a little confusing after discussing other amendments. Also, are organic resources = organic residues as you've referred to them before. Consistency in terminology would be good.

Four nutrient management treatments were selected for the present study, involving the application of three organic residues with varying qualities (at a rate of 4.0 t C ha$^{-1}$ yr$^{-1}$) and a control treatment (Control; no organic residues were added). These application rates are higher than those typically used by local farmers, which allows for a more pronounced assessment of the potential impacts on SOC. These organic residues were applied once annually before the long rainy season.

Concerning the justification for Thitonia, corn stover, and manure, it is already mentionned in the same paragraph and in table 1:
-   involving the application of three organic residues with **varying qualities** (at a rate of 4.0 t C ha−1 yr−1) and a control treatment (Control; no organic residues were added).
-   The organic resources utilized in the present study exhibited variations in quality, characterized by differences in nitrogen, lignin, and polyphenol contents. These included pruned leaves, including stems with a thickness of less than 2 cm, from Tithonia diversifolia (Tithonia; high quality with rapid turnover), stover of Zea mays (Stover; low quality and fast turnover), and locally available farmyard manure (Manure; intermediate to high quality with intermediate turnover). The relevant properties of the different organic inputs are given in (Table 1).

Table 1: Mean dry matter properties of the organic resources applied (source: Laub et al., 2023b)

| Measured property | Manure | Stover | Tithonia |
|---|---|---|---|
| C (g kg$^{-1}$) | 234 | 397 | 345 |
| N (g kg$^{-1}$) | 18.1 | 7.2 | 33.2 |
| C:N ratio | 12.3 | 58.7 | 12.4 |
| P (g kg$^{-1}$) | 3.1 | 0.4 | 2.3 |
| K (g kg$^{-1}$) | 19.4 | 9 | 37.2 |
| Lignin (g kg$^{-1}$) | 198 | 48 | 90 |
| Polyphenols (g kg$^{-1}$) | 7.8 | 11.3 | 19 |
| Lignin/N ratio | 6.9 | 6.2 | 2.6 |
| Kg N in 4.0 t C ha$^{-1}$,-N [+N] | 324 [564] | 68 [308] | 323 [563] |

Para at 146 is not needed.

We acknowledge that the paragraph at line 146 may not be strictly necessary. However, based on feedback received previously, we have found that the names of the different treatments were sometimes unclear to readers. Therefore, we prefer to retain this paragraph to ensure that the treatment names are clearly referenced if needed.

171: no need to repeat. Decide where and how you want to organize without repetition from above paragraph.

The first sentence and sentences that were repeating previous paragraph were removed and the second sentence added to the previous paragraph:

 Given the auger's 50 cm length, samples were obtained in two stages: the first sample down to 50 cm, followed by a second sample from 50 to 70 cm using an extension for the auger.

182: HCl DID or DID NOT show a reaction? If there is a reaction, I would think this means there ARE carbonates?
Samples were not treated with HCl before the analysis of OC because the pH of all samples was ≤ 6 and none of the 12 samples on which it was tested showed any reaction with HCl, indicating the absence of carbonates.

205-207: Extra line?
It was removed

256: "THis was only the case for...: and 267 "The results indicated..." should be in results, not methods.
It is an observation that impacts the way we interpret the results according to our methodolgy, therefore we argue that it is should rather be in the method than in the result section.

273-274:Cohen's d- "d" should be ital. "Significance" not "Significant"
The analysis was performed twice and both times the *Cohen's d* effect size measure (i.e., standardized difference between two means) was used as an input, along with a significance level of α = 0.05.

287: MIneral N fertilizer- spelling inconsistency
This was imporved in the new version of the manuscript: all sentences where mineral fertilizer was used instead of mineral N fertilizer were corrected and mineral N fertilizer was consitently used throughout the document.

Figure 3: the overlapping error bars make it pretty hard to read... any ways to clean this up or make it more legible would be good.
We acknowledge that this figure contains a lot of information and requires the reader to take time to understand it. However, after trying multiple ways to display it, we believe that this version is the most satisfactory as it effectively summarizes all the information. We do not agree that the overlapping error bars decrease the readability of the information it contains

387: Unclear when you are/aren't italicizing
Here Stover and Manure it should indeed be in italic. The rule we follow is that it is in italic (and with capital letter) when we speak about one of the exact treatment in our experiment, and it is not in italic when we speak about applying one of these OR in general.

392: Given the immediately preceding statement on your power analysis, I wonder if you could rephrase here to bring your conclusion in better alignment with your actual statistical tests.
Our study shows that, although statistical power indicated that we would need more samples to better detect down to which depth the different organic residues affect SOC, we were able to detect that the application of manure affects SOC stocks in the subsoil in a tropical long-term field trial.

449: "Very" would be better than "much responsive"
This does not apply at our study site in Embu, as the yield was not very responsive to N fertilization and crop residues were removed from the field after harvest (Laub et al., 2023a).

476-481: "Therefore, although there might be..." This sentence is too long- break it up. I'm also a bit weary of the sudden introduction of the co2 sequestration piece. Might be clearer to just keep it to your context of increasing SOC to offset losses in this cropping system. Some statement on whether you think there's any ways to actually

stabilize or rebuild SOC such that there isn't continual losses would be interesting to me as well... how much of this is the continual maize?

~~Therefore, although there might be little potential to effectively reduce atmospheric CO2 concentration by increasing SOC storage in soils with mainly low activity clays that were recently converted to agriculture, there is a potential to optimize agricultural nutrient management to minimize SOC losses after land use conversion from forest to agriculture and thereby forming a carbon sink compared to business as usual scenarios (i.e. low input of mineral and organic resources).~~

Therefore, although the carbon storage potential in soils composed mainly of low activity clay and recently converted to agriculture is low, it is possible to optimize agricultural nutrient management to minimize SOC losses after conversion of land use from forest to agriculture. Furthermore, this could be combined to other appropriate agricultural practices, such as increasing crop diversity and rotating crops, which have the potential to improve carbon stocks and positively impact yields (Yang et al., 2024). Combining these practices with the application of manure could lead to the tabilization of SOC stocks over time.

**Reviewer 2**

Dear Authors, thank you for the opportunity to review your interesting manuscript. I hope my recommendations prove helpful in improving the quality of your work. I have attached detailed comments in the PDF for some major and minor revisions.

Overall, the paper is well-written, but there are several instances where shorter, more concise phrasing could enhance readability. The figures and tables generally complement the findings. However, many appear unnecessary for addressing the proposed research questions. For instance, Fig 1 and Fig 4 were neither introduced in the introduction nor used to discuss the findings, raising questions about the rationale for their inclusion.

I found the focus of the text to be inconsistent, oscillating between topics on the subsoil, the entire soil profile, and comparisons of the top and subsoil. For instance, the title emphasizes depth-specific analysis, yet the abstract lean toward only subsoil dynamics, leading to some confusion. Moreover, the inclusion of mineral N fertilization in an experiment centered on SOC stocks was inadequately justified, even if such N fertilization didn't seem to impact the SOC stocks. I find the introduction to be the most troubling part of the paper. The argumentation is often meandering, with unclear research gaps and no compelling new directions this this study seeks to address. Here also, C isotopes are introduced abruptly, without sufficient context or connection to the broader question of subsoil C dynamics.

The part on the isotop was better integrated with importance for subsoil C dynamics

The questions are rather weak and fail to promise any novel or relevant insights, yet they remained only partly answered. Question 1 on how do organic residue application and mineral nitrogen fertilization affect SOC stock along a soil profile, was not answered. It seems self-evident, even without data, that organic residue application and mineral N fertilization would influence SOC stocks as was concluded in this study. I propose to rephrase the original questions into hypothesis-driven statements based on underlying mechanisms and processes to be investigated.

The research questions were merged into one clear question: "Down to which depth do the different organic residues affect SOC stocks"

Unfortunately, while the study a range of interesting soil analyses, the paper remains too superficial, offerring limited insights and failing to highlight exciting research directions. In addition to my PDF comments, I have also made some detailed comments below.

Abstract

The abstract proposes a study on SOC dynamics in the subsoil, yet, the boundary of the subsoil remained vague and largely undefined throughout the abstract. The research problem was also not mentioned, despite using the space to express the findings hat manure had the greatest effect on SOC at deeper depths. Nothing was mentioned about C isotopes and key soil properties such as pH and CECeff although they seem to be key to the findings of this study. This raises the question of whether measuring C isotopes and all the other soil properties were justified for this study. I suggest to use the first 3 lines of the abstract to indicate the research problem succinctly, and report on all measured parameters so that the reader can have a complete overview of what is to be expected in the paper.

Introduction
The paragraphs in the introduction lack coherence and fail to provide an adequate background to clarify the actual scientific problem. The initial two paragraphs primarily restate textbook knowledge, emphasizing the importance of SOC stocks for maintaining productive soils. While this is a fundamental concept, devoting two whole paragraphs to reiterate this point is an inefficient use of space.

An attempt to address the research problem appears in the third paragraph, but it concludes without proposing any exciting new research directions or unresolved questions. For a study focused on subsoil dynamics, depth-specific differences in soil nutrient dynamics are only introduced in the fourth paragraph. Even here, no explanation is offered regarding the specific processes dominating subsoils that might influence topsoil functions or plant nutrient acquisition. While the statement that 50% of SOC stocks lie below 30 cm is noteworthy, it lacks context—such stocks may be significant for long-term carbon storage but could be irrelevant for plant nutrition. Overall, the introduction fails to convincingly establish the novelty of the study. The proposed research questions appear to have already been addressed in standard soil science textbooks, making it difficult to discern what new insights this study aims to contribute.

Methods
Including a map showing the location of the study area in central Kenya would greatly assist readers unfamiliar with the geography, providing a quick and clear overview.

Section 2.1.1 is presented as though readers already have detailed prior knowledge of the experimental setup, including the findings of Laub et al. (2023b). This creates the impression that the results of this study are merely a subset of data from Laub et al., leaving little room for novelty or original contribution as it seems everything significant has already been reported in the prior work.
A key concern with the experimental design is the potential for unconstrained parameters to influence the results. For instance, while P and K are reportedly evenly supplied across treatments, N is included as a treatment variable (±N). This setup could lead to cascading effects that influence the reported findings, but this critical aspect is largely unexplained. This raises doubts about whether the experimental design is adequately optimized to address the research questions posed.
The statistical analysis is somewhat unclear and inconsistent. The mixed models approach to account for site-specific variability is reasonable, but it might be worth considering whether a generalized linear mixed model (GLMM) would be a better choice, given the diverse data types among the covariates (numeric and categorical). As there were no violation of any assumption made for performing a linear mixed model, we argure that it is not necessary to use a more complex model.

Moreover, it is unclear whether all statistical assumptions were verified on the residuals or the raw data. All statistical assumptions were indeed verified on the residuals of the model,

A fundamental question remains: on what theoretical basis were the selected covariates expected to influence SOC stocks? The paper does not clarify whether a stepwise selection process was employed to exclude less explanatory variables or whether the model is purely theoretical. For instance, the removal of soil pH and CEC from the model specifications is unexplained. This is not correct, reason for removal of soil pH and CEC was explained in the method section of the linear mixed model (l.235):
"Other measured parameters such as pH and CECeff were not considered in this model, because they correlated with the application of all three types of organic residues (especially manure) (Fig. 1)."

It is of course inadequate to simply state that they had a strong relationship with the SOC stocks. We did not simply state that they have strong relationship with the SOC stocks (see above). While it is true that soil pH and CEC have a strong relationship with SOC stocks, it is important to note that these parameters are also influenced by the addition of organic residues (OR) to the soil. Since the focus of the study is on the impact of organic residues, we retained the factor "organic residue" in the model. By doing so, we aimed to isolate the effect of organic residues on SOC stocks without the confounding influence of soil pH and CEC, which are themselves affected by the organic residue application.

Additionally, how soil depth is treated in the model—whether as ordered categories or independent variables—has significant implications for interpreting the outcomes. Clarification on this point is essential. Soil depth was included as a continuous numerical variable in the linear mixed model using the lmer function. This approach allows us to capture the continuous nature of depth and its potential linear relationship with the dependent variable.

In the end, how model parameters were validated and where necessary fine-tuned was not reported.
This is a very valid point that you raised here, and the fine-tuning of the model was now performed based on the detailed and constructive inputs provided by Reviewer 1:

Method
To avoid overfitting and ensure the robustness of our findings, we excluded interactions. This approach simplifies the model, focusing on the main effects and accounting for variability through random effects, thereby reducing the risk of capturing noise rather than true underlying patterns. We evaluated the robustness of our model by fitting six different linear mixed-effects models using the lmer function from the lme4 package in R. Each model included OC stocks as the response variable, the sampling plot as random effect and different combinations of fixed effect predictor variables: model 1 (depth, organic residues, mineral fertilizer, silt content, silt content), model 2 (depth, silt content, silt content), model 3 (depth, organic residues), model 4 (depth, mineral fertilizer), model 5 (organic fertilizer), model 6 (mineral fertilizer). We then compared these models with an ANOVA to determine the best-fitting model based on AIC, BIC, log-likelihood, and deviance values. To further test the effect of organic residue treatment application in the deeper soil layer, the same analysis was performed on a subset containing only subsoil layers. The R-squared, fixed effect

Results:
According to the ANOVA performed on all the different models, the model including only depth and had the lowest AIC (596.05) and BIC (618.56), the highest log-likelihood (-291.03) and the lowest deviance (582.05), suggesting it was the best-fitting model among the four (Supplementaty table 1). The $R^2$ of this model was 76.3%, representing the percentage of the variance in carbon levels that this model explained, with 64.9% attributed to fixed effects and 11.4% to random effects. According to the model's result, depth, application of manure and application of maize stover were significantly impacting OC stocks, not the application of tithonia (Table 2). The same analysis performed on a subset conatining only subsoil data yileded very similar results with the best fitting model including only depth and organic residues as fixed effect and having a $R^2$ of 0.64. According to the subsoil best fit model, beside depth, only manure was significantly affecting OC stocks (Supplementary table 2). The application of mineral fertilizer was never considered to significantly affect OC stocks by any model.

Table 2: Summary statistics of the linear mixed effect model for soil organic carbon stocks on all measured depth layers

|  | estimate | Standard error | p value | |
| --- | --- | --- | --- | --- |
| (Intercept) | 8.98 | 0.41 | < 2e-16 | * |
| Depth | -0.07 | 0.00 | < 2e-16 | * |
| Manure | 2.04 | 0.53 | 0.001 | * |
| Stover | 1.59 | 0.53 | 0.007 | * |
| Tithonia | 0.92 | 0.53 | 0.095 | |

Supplementary table 1: ANOVA Comparison of Linear Mixed-Effects Models performed on all meaured depth layers

| Model | npar | AIC | BIC | logLik | deviance | Chisq | Df | Pr(>Chisq) |
| --- | --- | --- | --- | --- | --- | --- | --- | --- |
| C ~ Mineral Fertilizer + (1 \| Plot) | 4 | 802.2 | 815.06 | -397.1 | 794.2 | | | |
| C ~ Depth + Mineral Fertilizer + (1 \| Plot) | 5 | 605.18 | 621.25 | -297.59 | 595.18 | 199.02 | 1 | <2e-16 *** |
| C ~ Depth + silt + sand + (1 \| Plot) | 6 | 608.27 | 627.56 | -298.13 | 596.27 | 0 | 1 | 1 |

| | | | | | | | | |
|---|---|---|---|---|---|---|---|---|
| C ~ Organic Residue + (1 \| Plot) | 6 | 793.07 | 812.36 | -390.54 | 781.07 | 0 | 0 | |
| C ~ Depth + Organic Residue + (1 \| Plot) | 7 | 596.05 | 618.56 | -291.03 | 582.05 | 199.02 | 1 | <2e-16 *** |
| C ~ Depth + Organic Residue + silt + sand + (1 \| Plot) | 10 | 599.69 | 631.84 | -289.85 | 579.69 | 2.36 | 3 | 0.5013 |

Supplementary table 2: Summary statistics of the linear mixed effect model for soil organic carbon stocks on subsoil depth layers

| | estimate | Standard error | p value | |
|---|---|---|---|---|
| (Intercept) | 9.17 | 0.67 | < 2e-16 | * |
| Depth | -0.07 | 0.01 | 5.36e-10 | * |
| Manure | 1.86 | 0.61 | 0.007 | * |
| Stover | 1.03 | 0.61 | 0.11 | |
| Tithonia | 0.40 | 0.61 | 0.52 | |

The use of a t-test over an ANOVA for comparing treatments and controls is particularly concerning. This choice suggests a lack of clarity on how to handle the soil depth variable, which further complicates interpretation. It is well known that multiple t-tests increase the risk of Type I errors (false positives), yet this issue is not addressed.

We acknowledge the concern of reviewer 2 regarding type I errors with multiple t-tests. However, our study is hypothesis-driven, and we observe a clear trend: significant differences observed in the top layer diminish with depth. As we analyze deeper soil layers, the observed differences between OR treatments and the control decrease. We perform this analysis layer by layer to determine down to which depth these statistical differences are observed. We are not randomly testing a large number of treatmnents against each other (in which case the multiple t-tests would indeed increase the risk of type I errors). Given the limited data (due to sampling in a long-term field trial) and increasing variability with depth, we chose t-tests over an Tukey's test because an Tukey's test would be even more affected by the limited statistical power. When statistical significance was not achieved, the power was below 80%, supporting our choice. Therefore, we argue that the typical type I error inflation does not apply in our context and that we are more likely to experience type II error. Neverthelss, we also performed an ANOVA and provided the results in the new version of the manuscript.

The following paragraph was added to the statistical methodology section:
Due to violation of the homogeneity of the variance over some depth layers, OC stock data were log transformed. An ANOVA was then performed on each depth layer, after normality and homogeneity assumptions of the log-transformed data were confirmed (Supplementary table 5). Given the limited data from a long-term field trial and increasing variability with depth, we opted for t-tests over Tukey's test to identify which OR treatment had significantly higher OC stocks than the control treatment and at which depth layers. Performing Tukey's test on the ANOVA results would be more affected by limited statistical power. Our choice was supported by the fact that when statistical significance of t-test was not achieved, the power was below always 80% (Figure 4). Therefore, we argue that type I error inflation due to the amount of t-tests performed does not apply in our context, and we are more likely to encounter type II errors.

Moreover, the assumptions underlying the t-tests were neither tested nor reported.
This was now made available in the suppelementary information

Given these shortcomings, the justification for conducting a power analysis is unclear. I strongly recommend a thorough re-evaluation of the statistical methods to ensure that each test is theoretically grounded and suitable for combining data from different treatments. This re-evaluation should include a clear explanation of why certain variables were included or excluded and how soil depth was treated within the model.

The result of the ANOVA has been added, as you requested in your detailed comments on the pdf document. However, we compared each organic residue treatment with the control using a t-test. The reasons for this are explained in detail in the manuscript.

Results
The results section does not adequately address the research questions and instead shifts focus to soil properties that were not introduced earlier in the paper. For example, it is unclear why Section 3.1 is included, particularly as it is not integrated into the discussion.
Results from Section 3.1 were integrated in the discussion of the new version of the manuscript.

While these soil properties may be important, they are not referenced in the abstract or introduction, raising questions about their relevance and utility in opening the results section.
The use of vague phrases such as "non-significant but... the tendency" is problematic, as statistical outputs are binary (significant or not significant). Such wording undermines the credibility of the statistical interpretation. Although Fig. 1 highlights some interesting trends in soil pH and CECeff, these trends are largely ignored in the results section. For example, while manure treatment shows no change in soil pH across depth layers, there is a clear decreasing trend in CECeff with depth. Conversely, for the control, as well as the stover and Tithonia treatments, soil pH shows a somewhat increasing trend with depth, while CECeff remains relatively stable. What mechanisms might explain these contrasting patterns? This is a missed opportunity to link observed trends with meaningful interpretations.

Furthermore, the entirety of Fig. 4 could be removed or moved to the supplementary material without any loss of key information.
This is a crucial aspect of our study, justifying the use of a simple t-test rather than more complex statistical tests and of combining the ±N treatments. It also highlights an essential point for future research focusing on subsoil. This was further highlighted in the new version of the manuscript.

Discussion
A major premise in the discussion is that adding organic residues under the various treatments limits SOC loss, as these treatments consistently showed higher SOC stocks compared to the control. This argument is perplexing because "SOC losses" are neither defined nor measured as part of the experiment.
These SOC loss were identified by the work of Laub et al. 2023, this was clearly defined in the study description and there was a plot of these losses over year avalailbe in the supplementary (Figure S8).

Moreover, instead of offering a logical interpretation of their own results, the authors rely heavily on finding confirmation in other studies. This approach weakens the discussion, as it fails to provide plausible explanations for the observed outcomes. Consequently, the discussion lacks depth, giving the impression of a superficial analysis that does not fully explore the implications or mechanisms underlying the findings.
The study design does not allow for conclusions on the mechanistic aspects of the OC cycle. However, our data suggest that agricultural practices impact the soil environment at greater depths than typically studied. This raises the question of whether the standard sampling depth threshold of 25-30 cm should be reconsidered. Our findings indicate that deeper soil layers may be significantly affected, warranting further investigation into the implications of these practices beyond the conventional depth and raising the question of the importance of the subsoil for crop productivity.

Conclusion
One would expect the two questions posed in the introduction to be revisited here, reflecting on how the findings contribute to a broader understanding of SOC stocks in the subsoil. However, this was not the case. Instead, the speculations presented in the discussion were validated as final conclusions without critically examining the insights provided by the results.

---

## Referee Report (RR1)

**This manuscript is much improved! Well done. Many of my comments are somewhat stylistic, and are just suggestions for further honing the MS before publication. Primarily, I'd love to get more context on why you think manure had this effect, and citing relevant literature to this effect. I'd also like authors to consider including a bit more context on the trajectory of land-use change in the region to contextualize deforestation, and the results presented.**

- I think your overall finding in the abstract could still be be a bit clearer and to the point - Overall our findings indicate that SOC in the subsoil… was only impacted by a single type of organic amendment, manure, while other amendments only increased surface SOC. —> Could nest some of the importance of the study in referencing the global importance of the subsoil as a C-reservoir that is under appreciated and that could be impacted by management decisions.
- WC "ever-more" is slightly strange. I understand the point but I think the first line could be streamlined. "Satisfy the food demand of an increasingly demanding and growing population." I also sometimes caution this as the introduction when studies have shown that, calorically, food production is actually sufficient for our current and growing populations, and that food and resource distribution and access are primary forces in solving global hunger.
- Citation would be good for SOC functional properties in 55-57.
- Could streamline introduction of Organic residues by replacing "organic materials" in line 64 with OR to avoid parenthetical.
- Transition to paragraph at line 75 could be smoother. Transition with OR first and then can mention SOC turnover (flip the sentence).
- Replace "its" in line 76 with OR's
- Is t"the amount of SOC" the stock?
- Do you have to use (lignin + Polyphenol) Could this just be defined as normal C:N ratios?
- In general, "SOC content" feels a little vague - could you clarify whether it is concentration, or stocks in the various studies that you refer to throughout.
- The second to last line (Different factors… line 89) feels like it deviates from the setup of your final line. Think through the logical flow of this paragraph a bit more to better articulate the research gap that you are filing - I think this is the need for more research on OR in SSA and tropical systems,
- Transition to 95, maybe add "additionally" to start.
- Not sure where c3, c4 fits in to the picture of the introduction. I think the introduction of 14C for aging soil C is much improved, but I don't think the last line on improving models fits in the intro either… this can be saved for discussion.
- Are nutrient management treatments from Laub 2023b the same as the ones you have? Be clearer, and the third sentence feels repetitive. The other field sites are not relevant either in 2nd sentence.
- Delete "at" in 141.
- I appreciate the effort to maintain figure y-axes in 2, but I wonder if it'd be clearer to have a slightly adjusted y-axis for the right-most plots to maybe may at ~100? and perhaps reconfigure so that the differences can be seen a bit more clearly - not urgent but if possible.

- Re-reference the model number assigned in the methods for the best-fit model in lines 505-506
- Line 515 clunky - re-write. "According to the best-fit model for subsoils, manure and depth were the only significant variables affecting OC stocks" or something.
- Line 521-522 "The effect... on SOC stocks not needed here.
- Omit "the deepest" and shorten - "Manure significantly impacted OC stocks down to 60cm" is more concise
- Unsure of which residue treatment is considered in table 3 - all three?
- "This means that after 38 growing seasons over 19 years, and considering the top 70 cm of the soil, Manure-N, as the most effective treatment to limit SOC losses, could maintain 38.5 ± 8.8 t OC ha-1 more SOC compared to the least efficient treatment, Control+N." A little hard to follow with all the clauses, and also maybe should go into the discussion rather than results
- In the results, you switch to discussion of C/N whereas before you use lignin/polyphenol. I know these are similar concepts but I think uniformity or at least clarifying the transition would be good.
- 617, I might add a "statistically" significant. I think you do a nice job of showing that there are likely effects, but that for greater statistical conidence, you require more samples... It could also be interesting to discuss the impact that this might have on sampling schemes in general... as many deep samples pose significant logistical challenges
- Line 652: it is at the highest end of typical manure application rates, ranging from... Not sure how the following line actually logically relates... on the impact of manure may not be as gratifying when considering non-longterm trials.
- When you discuss the mitigation in losses of SOC, it would be interesting to have you discuss some of the mechanisms that might be at play - while you cannot make definite statements with your study design, I suspect that there could be work that supports certain hypotheses to support your observations with each of your treatments, and in particular with manure. What is it about manure (C/N ratio, something else) that allows It to make such permeating differences relative to the other treatments? I think something to consider, which is not mentioned, is the fact that DOC and leachates from manure may percolate further down than plant-based leachates? Curious to know what you think is at play here in your system.
- Not "very" responsive, rather than "much responsive" in line 708.
- Is there increasing deforestation in the area - its something briefly mentioned as the site condition, but having some sense of the broader context of deforestation in your system may add some nice detail for you concluding thoughts, given that the idea of deforestation and land change only come up in passing once, much earlier in the paper.

---

## Author Response (AR2)

Dear Reviewer,

Thank you for your detailed and constructive feedback on our manuscript. Your suggestions will significantly improve the readability and clarity of our paper. Our response to each comment are written in blue below the corresponding comment and the sentence that were adapted are italicized. We appreciate the time and effort you invested in reviewing our work. Thank you very much.
* * *
**This manuscript is much improved! Well done. Many of my comments are somewhat stylistic, and are just suggestions for further honing the MS before publication. Primarily, I'd love to get more context on why you think manure had this effect, and citing relevant literature to this effect. I'd also like authors to consider including a bit more context on the trajectory of land-use change in the region to contextualize deforestation, and the results presented.**

The following sentence was added in the introduction:
*Eastern Africa followed this trend, experiencing a substantial land use transformation between 1988 and 2017, with cropland expanding by 35% and settlements by 43%, primarily at the expense of woody vegetation*

- I think your overall finding in the abstract could still be be a bit clearer and to the point - Overall our findings indicate that SOC in the subsoil… was only impacted by a single type of organic amendment, manure, while other amendments only increased surface SOC. —> Could nest some of the importance of the study in referencing the global importance of the subsoil as a C-reservoir that is under appreciated and that could be impacted by management decisions.

The sentence you are referring to in the abstract was modified as follows:
*Our findings demonstrate that SOC in the subsoil comprised 48.5% ± 1.7% of the total SOC stocks across the 0-70 cm soil profile, yet only manure application affected subsoil OC levels, while other organic amendments only increased SOC in the surface layer.*

- WC "ever-more" is slightly strange. I understand the point but I think the first line could be streamlined. "Satisfy the food demand of an increasingly demanding and growing population." I also sometimes caution this as the introduction when studies have shown that, calorically, food production is actually sufficient for our current and growing populations, and that food and resource distribution and access are primary forces in solving global hunger.

The first line was adapted as suggested:
*Globally, food production systems need to satisfy food demand of an increasingly demanding and growing population.*

- Citation would be good for SOC functional properties in 55-57.

A citation was included:
*Therefore, agricultural management should not only aim at increasing crop yield in the short*

*term, but also at maintaining SOC at a level that maintains the key functions that it regulates (e.g., nutrient cycling, water retention, improved cation exchange capacity (CEC), soil structure stabilization, promotion of biodiversity) in the long term (Wiesmeier et al., 2019).*

-       Could streamline introduction of Organic residues by replacing "organic materials" in line 64 with OR to avoid parenthetical.

The text was adapted as suggested:
*This approach promotes various practices, including the combined application of mineral fertilizer and organic residues (OR) such as plant residues, manure, and compost.*

-       Transition to paragraph at line 75 could be smoother. Transition with OR first and then can mention SOC turnover (flip the sentence).

The text was adapted as follows:
*Applying OR over multiple years in tropical croplands can significantly increase the SOC content (Fujisaki et al., 2018), despite the faster SOC turnover rates in tropical compared to temperate climates (Wang et al., 2018).*

-       Replace "its" in line 76 with OR's

The text was adapted as suggested:
*However, the contribution of OR to the amount of SOC is, in part, controlled by the type of added OR (Córdova et al., 2018).*

-       Is t"the amount of SOC" the stock?

No, "It" is the OR.
The "It's" contribution was replaced by "*the contribution of OR*" in the text.

-       Do you have to use (lignin + Polyphenol) Could this just be defined as normal C:N ratios?

We chose to use the definition of organic residues as it was initially defined for this long-term field trial and accordingly to the definition of ISFM by Vanlauwe et al. (2005). It is relevant to discern the composition of *Tithonia* and *Manure* which have similar C:N ratio, but different (lignin+polyphenol)/N ratio (Table 1).

-       In general, "SOC content" feels a little vague - could you clarify whether it is concentration, or stocks in the various studies that you refer to throughout.

We agree that SOC content is less precise than stocks or concentration. Therefore, we now made sure that when we were discussing about our own data or directly comparing amounts of SOC with other studies, we always used the appropriate unit. We kept the use of content for some cases where we were referring to a mix of studies that were not using the same unit.

-       The second to last line (Different factors... line 89) feels like it deviates from the setup of your final line. Think through the logical flow of this paragraph a bit more to better articulate the research gap that you are filing - I think this is the need for more research on OR in SSA and tropical systems,

The paragraph was adapted as follows:
*The contradictory findings across tropical regions highlight the complexity of SOC dynamics in these agroecosystems where different factors, such as initially high SOC contents, favorable conditions to decomposition, and the limited capacity of 1:1 kaolinite clay minerals to stabilize OC, contribute to consistent SOC losses despite the application of organic residues in tropical soils (Laub et al., 2023b; Six et al., 2002; Sommer et al., 2018 While some studies demonstrate increases in SOC stocks through management practices, others report continued SOC losses even with substantial organic inputs, suggesting that local soil properties and environmental conditions play crucial roles. This knowledge gap is particularly evident in tropical agroecosystems, where the interactions between organic amendments, soil properties, and SOC stabilization mechanisms remain poorly understood, necessitating further research.*

-       Transition to 95, maybe add "additionally" to start.

The transition was improved as suggested:
Additionally, while field studies on plant nutrient acquisition from the subsoil are rare, ...

-       Not sure where c3, c4 fits in to the picture of the introduction. I think the introduction of 14C for aging soil C is much improved, but I don't think the last line on improving models fits in the intro either... this can be saved for discussion.

This was adapted as suggested:
*Also $\delta^{13}C$ can be used to detect the portion of C3 compared to C4 plant residues in the subsoil, enabling for example the tracking of maize-derived carbon inputs (Balesdent et al., 1987; Farquhar G D et al., 1989).*

-       Are nutrient management treatments from Laub 2023b the same as the ones you have? Be clearer, and the third sentence feels repetitive. The other field sites are not relevant either in 2nd sentence.

The text was adapted as follows:
*In a recent study conducted at the same long-term field studied in the present manuscript (Embu), Laub et al. (2023b) showed that all nutrient management treatments resulted in significant topsoil SOC losses, while the use of farmyard manure was the most effective strategy to minimize the loss of SOC.*

- Delete "at" in 141.

The "*at*" was remved, as suggested.

- I appreciate the effort to maintain figure y-axes in 2, but I wonder if it'd be clearer to have a slightly adjusted y-axis for the right-most plots to maybe may at ~100? and perhaps reconfigure so that the differences can be seen a bit more clearly - not urgent but if possible.

We are not sure to understand what how exactly you "mean by slightly adjust […] at ~100?" What is important to us with this plot is that it shows that about 50% of the SOC of the 0-70 cm profile is in the 30-70 cm and to highlight that at this depth only the manure treatment has a significantly higher SOC than the control.

- Re-reference the model number assigned in the methods for the best-fit model in lines 505-506

The text was adapted as suggested:
*According to the ANOVA performed on all the different statistical models, the model including all depth and the OR (i.e., model 1) had …*

- Line 515 clunky - re-write. "According to the best-fit model for subsoils, manure and depth were the only significant variables affecting OC stocks" or something.

The text was adapted as suggested:
*According to the best-fit model for subsoils, manure and depth were the only significant variables affecting OC stocks (Supplementary Table 2).*

- Line 521-522 "The effect… on SOC stocks not needed here.

It was not clear to us what this comment is referring to.

- Omit "the deepest" and shorten - "Manure significantly impacted OC stocks down to 60cm" is more concise

This was adapted exactly as suggested

- Unsure of which residue treatment is considered in table 3 - all three?

Yes, all three. This was indeed not clear and the title of the table was modified as follows:
*Table 3: Summary statistics of ANOVA testing the effect of all OR treatments on OC stocks across depth layers*

- "This means that after 38 growing seasons over 19 years, and considering the top 70 cm of the soil, Manure-N, as the most effective treatment to limit SOC losses, could maintain 38.5 ± 8.8 t OC ha-1 more SOC compared to the least efficient treatment, Control+N." A little hard to follow with all the clauses, and also maybe should go into the discussion rather than results

The sentence was simplified as follows:
*Therefore, the Manure-N treatment maintained 38.5 ± 8.8 t OC ha-1 more SOC in the 0-70 cm depth layer compared to the Control+N treatment.*
- In the results, you switch to discussion of C/N whereas before you use lignin/polyphenol. I know these are similar concepts but I think uniformity or at least clarifying the transition would be good.

This is true. However, in the discussion, we consider the C/N ratio of the soil, while in the introduction of the concept, the lignin/polyphenol refers to the quality classification of the organic residues.

- 617, I might add a "statistically" significant. I think you do a nice job of showing that there are likely effects, but that for greater statistical confidence, you require more samples… It could also be interesting to discuss the impact that this might have on sampling schemes in general… as many deep samples pose significant logistical challenges

Statistically was added, exactly as suggested

- Line 652: it is at the highest end of typical manure application rates, ranging from… Not sure how the following line actually logically relates… on the impact of manure may not be as gratifying when considering non-longterm trials.

It was not very logical indeed. The sentence was modified as follows:
*Therefore, the impact of using manure may not be as great in the deeper layers when considering a scenario where common rate of manure are applied*

- When you discuss the mitigation in losses of SOC, it would be interesting to have you discuss some of the mechanisms that might be at play - while you cannot make definite statements with your study design, I suspect that there could be work that supports certain hypotheses to support your observations with each of your treatments, and in particular with manure. What is it about manure (C/N ratio, something else) that allows It to make such permeating differences relative to the other treatments? I think something to consider, which is not mentioned, is the fact that DOC and leachates from manure may percolate further down than plant-based leachates? Curious to know what you think is at play here in your system.

This sentence was added in the discussion
*A known mechanism for OC transport across soil profile is in the form of dissolved organic carbon (DOC), which moves downward through cycles of sorption and desorption (Kaiser and Kalbitz, 2012b; Uselman et al., 2007). In a long term in China, Liu et al. (2013) observed that manure applications increased the amount of DOC down to 60 cm as compared to straw residues application, supporting that manure may percolate further down than plant residues.*

-       Not "very" responsive, rather than "much responsive" in line 708.

This was modified exactly as suggested.

-       Is there increasing deforestation in the area - its something briefly mentioned as the site condition, but having some sense of the broader context of deforestation in your system may add some nice detail for you concluding thoughts, given that the idea of deforestation and land change only come up in passing once, much earlier in the paper.

A reference to this was added at the beginning of the introduction.

---

## Author Response (AR3)

Dear Editor,

When we tried writing it out the change exactly as suggested at line 111 which was:
*"In a recent study conducted at the same long term field trial studied in the present manuscript (Embu), Laub et al. (2023b) showed that all nutrient management treatments resulted in significant topsoil SOC losses, while the use of farmyard manure was the most effective strategy to minimize the loss of SOC."*, we obtained:

"In a recent study conducted at the same long term field trial studied as in the present manuscript (Embu)..."

This felt a bit awkward to read, so we tweaked it to:
"In a recent study conducted at the same long term field trial that is studied in the present manuscript (Embu)..."

We think this still captures what you were looking for while flowing a bit better. Hope this works.